# Bile canaliculi remodeling activates YAP via the actin cytoskeleton during liver regeneration

Kirstin Meyer[1], Hernan Morales-Navarrete[1] (iD), Sarah Seifert[1], Michaela Wilsch-Braeuninger[1], Uta Dahmen[2], Elly M Tanaka[3], Lutz Brusch[4], Yannis Kalaidzidis[1,5] & Marino Zerial[1,*] (iD)

## Abstract

The mechanisms of organ size control remain poorly understood. A key question is how cells collectively sense the overall status of a tissue. We addressed this problem focusing on mouse liver regeneration. Using digital tissue reconstruction and quantitative image analysis, we found that the apical surface of hepatocytes forming the bile canalicular network expands concomitant with an increase in F-actin and phospho-myosin, to compensate an overload of bile acids. These changes are sensed by the Hippo transcriptional co-activator YAP, which localizes to apical F-actin-rich regions and translocates to the nucleus in dependence of the integrity of the actin cytoskeleton. This mechanism tolerates moderate bile acid fluctuations under tissue homeostasis, but activates YAP in response to sustained bile acid overload. Using an integrated biophysical–biochemical model of bile pressure and Hippo signaling, we explained this behavior by the existence of a mechano-sensory mechanism that activates YAP in a switch-like manner. We propose that the apical surface of hepatocytes acts as a self-regulatory mechano-sensory system that responds to critical levels of bile acids as readout of tissue status.

**Keywords** actin cytoskeleton; bile canaliculi; liver regeneration; mechano-sensing; YAP

**Subject Categories** Cell Adhesion, Polarity & Cytoskeleton; Signal Transduction

**Mol Syst Biol. (2020) 16: e8985**

## Introduction

Organ size control is a fundamental aspect of morphogenesis. It requires a control system acting throughout scales of organization to coordinate local cell behavior with global tissue properties. Structural, biochemical, and mechanical properties of tissues regulate cell proliferation and differentiation (Nelson *et al*, 2005; Engler *et al*, 2006), cell geometry (Farhadifar *et al*, 2007), and the growth and organization of cells into tissues (Rauzi *et al*, 2008). Mechanical properties of tissues emerge from the interaction of cells and their environment. These include local cell–cell and cell–matrix interactions as well as systemic factors such as blood pressure, fluid shear stress, and muscle tension. Cells respond to biochemical or mechanical alterations through the activation of signaling pathways and downstream effectors, of which one of the most important is the actin cytoskeleton. It is able to generate forces via the acto-myosin system and controls cell behavior through signaling cascades, such as the Hippo and SRF pathways. Current concepts on the reciprocal regulation of tissue structure and cell behavior are primarily derived from studies *in vitro* and *ex vivo* (Allioux-Guérin *et al*, 2009; Connelly *et al*, 2010). However, the responses of cells to the structural and mechanical properties within tissues are largely unexplored.

Liver regeneration provides an excellent example of organ size control. Liver mass scales with body size, constituting about 5% of the body weight in rodents (Boxenbaum, 1980), and the organ has the capacity to regenerate the original mass (Higgins & Anderson, 1931). The liver consists of functional units or *lobuli*. Each lobule contains two opposing fluid networks, the sinusoidal endothelial and bile canaliculi (BC) networks that transport blood and bile between portal and central vein (PV-CV), respectively. The hepatocytes are polarized cells at the interface of both networks. They take up metabolites from the blood via their basal plasma membranes and secrete waste products and bile via their apical membranes, which, collectively, form a continuous network that drains into bile ducts.

Several signaling pathways drive liver regeneration (Huang *et al*, 2006; Natarajan *et al*, 2007), including the Hippo pathway (Loforese *et al*, 2017; Lu *et al*, 2018). The Hippo pathway is activated during liver regeneration (Grijalva *et al*, 2014) and primarily driven via the co-transcriptional activator yes-associated protein (YAP). The pathway responds to mechano-sensory signals from the actin cytoskeleton, cell polarity complexes, and cell–cell or cell–matrix junctions (Chen *et al*, 2010; Aragona *et al*, 2013; Yang *et al*, 2015) but also to

---

1 Max Planck Institute of Molecular Cell Biology and Genetics, Dresden, Germany
2 Experimental Transplantation Surgery, Department of General, Visceral and Vascular Surgery, Jena University Hospital, Jena, Germany
3 Research Institute of Molecular Pathology, Vienna BioCenter, Vienna, Austria
4 Center for Information Services and High Performance Computing, Technische Universität Dresden, Dresden, Germany
5 Faculty of Bioengineering and Bioinformatics, Moscow State University, Moscow, Russia
*Corresponding author. Tel: +49 351 210 1100; E-mail: zerial@mpi-cbg.de

high concentrations of bile acids (BA) (Anakk *et al*, 2013). Despite these molecular insights, how cells sense the overall tissue status and size remains an open question. In particular, it is unclear how cells integrate metabolic (BA levels) and/or mechanical alterations, such as changes in blood and bile pressure, to control liver size during homeostasis and regeneration. Here, we addressed these questions by applying high-resolution microscopy and quantitative 3D image analysis (Morales-Navarrete *et al*, 2015; Meyer *et al*, 2017) to explore tissue and cellular alterations during liver regeneration. Our results unravel a new link between liver metabolism and Hippo signaling where the signaling pathway is activated through the mechanical properties of the biliary network in a switch-like manner.

# Results

## Alterations of the BC network during liver regeneration

Tissue sections of liver from different time points after partial hepatectomy (PH; 0.8–5 days post-PH) were stained for the apical marker CD13 by immunofluorescence (IF), imaged at high resolution by confocal microscopy and the 3D BC network reconstructed from IF image stacks. The analysis revealed remarkable changes of BC network topology and geometry. The BC network dilated and branched as early as 0.8 day post-PH (Fig 1A) throughout the entire CV-PV axis and reversed to normal at ~ 3–5 days post-PH. Spatial quantification revealed an increase in BC diameter by up to 27% (zone 11) and on average by 21% throughout the entire CV-PV axis at 1.5 days after PH as compared to the untreated liver (1.96–2.27 μm in untreated liver; 2.40–2.61 μm at 1.5 days post-PH; Fig 1B). At 5 days post-PH, the BC diameter was still increased, but only by 11% on average (BC diameter at 5 days post-PH, 2.14–2.51 μm). In addition, the BC lacked the typical smooth and regular appearance at this resolution and acquired a rough surface texture (Fig 1A, compare BC in PV area of untreated vs. 1.5 days post-PH). This could reflect alterations of the acto-myosin system, which mediates apical contractility and regulates BC geometry and bile flow (Watanabe *et al*, 1991; Meyer *et al*, 2017). Interestingly, the observed structural changes preceded the reported temporal profile of hepatocyte proliferation, which starts at 1.5 days post-PH and peaks at 2 days (Zou *et al*, 2012). The structural alterations of the BC network may just be an epiphenomenon or play an active role in liver regeneration. Therefore, we set to analyze the nature and cause of BC network expansion.

## Expansion of the apical surface of hepatocytes during liver regeneration

The observed BC diameter increase could be due to a bona fide expansion of the apical surface area of hepatocytes or be only apparent, e.g., due to decreased contractility of the acto-myosin cortex and/or flattening of microvilli. To distinguish between these possibilities, we quantified BC membrane length and perimeter by electron microscopy (EM) at 1.8 days post-PH or control-operated (sham) and untreated mice (Fig 1C–E). The apical membrane length was determined by segmentation of the BC, whereas the BC perimeter was estimated by calculating the minimal enclosing ellipse of

these segmentations (Appendix Fig S1). The EM analysis (Fig 1C) confirmed the BC expansion (Fig 1A). Both BC perimeter and total membrane length increased to a similar extent, 1.5 and 1.4-fold, respectively (Fig 1D and E). Thus, the primary cause of network dilation is an increase in total apical membrane of hepatocytes.

Liver resection induces a transient overload of BA (Péan *et al*, 2013). The observed BC network expansion could be a compensatory response to increased apical secretion and changes of biliary fluid dynamics. To test whether BA overload can replicate the BC network expansion, we examined the BC network structure in cholestatic livers, using the model of bile duct ligation (BDL). In this model, the common bile duct is ligated, resulting in BA overload of the organ. After BDL, the BC network in the liver was branched and expanded (Fig 1F), strikingly resembling regenerating liver. As in regeneration, the expansion occurred already at 1 day post-BDL throughout the entire CV-PV axis, preceding hepatocyte proliferation (Georgiev *et al*, 2008). Based on these results, we hypothesize that the expansion of the BC network could be part of a mechano-sensory system that responds to increased BA levels to induce liver growth during regeneration.

## Expansion of the apical surface of hepatocytes is associated with increased acto-myosin levels

Bile canaliculi possess a dense sub-apical contractile actin mesh via the acto-myosin system (Watanabe *et al*, 1991; Meyer *et al*, 2017) which contributes to BC geometry and bile flow (Meyer *et al*, 2017). To test whether the actin cytoskeleton is modified along with the changes in BC during regeneration, we quantified the levels of apical F-actin (phalloidin, Fig 2A, Appendix Fig S2A) and phospho-Myosin light chain (pMLC, Fig 2C, Appendix Fig S2B) at the apical area of hepatocytes at different time points after PH or sham operation. First, we had to take into consideration the effects of the sham operation itself on the actin cytoskeleton: The apical F-actin density progressively decreased by up to $37 \pm 9\%$ (mean $\pm$ s.e.m.) compared to untreated mice (Fig 2B), presumably due to the laparotomy and/or anesthesia/analgesia. However, upon PH, the F-actin intensity was higher than after sham operation and fluctuated at baseline levels as compared to untreated mice, with a maximum increase of $17 \pm 10\%$ (mean $\pm$ s.e.m) at 1.5 days (Fig 2B). In comparison with F-actin, apical pMLC intensity increased more dramatically during regeneration (Fig 2C and D). Whereas, in sham-operated mice, apical pMLC levels remained similar to baseline levels, within a maximum increase of $30 \pm 9\%$ (mean $\pm$ s.e.m.) and decrease of $13 \pm 11\%$ (Fig 2D, Appendix Fig S2B), upon PH, they raised early and remained elevated by about 70% until ~ 2.8 days post-PH as compared to untreated mice (Fig 2D). Overall, the expansion of the BC network during regeneration is associated with a concomitant increase in apical F-actin and pMLC levels, i.e., signs of increased acto-myosin contractility.

## BC network expansion correlates with YAP activation and proliferation during regeneration

The actin cytoskeleton converts mechanical forces into biochemical signals. Since the Hippo pathway is a prominent mechano-sensor downstream of the actin cytoskeleton and can be activated by BA (Anakk *et al*, 2013), we chose YAP/TAZ as a marker for a signaling

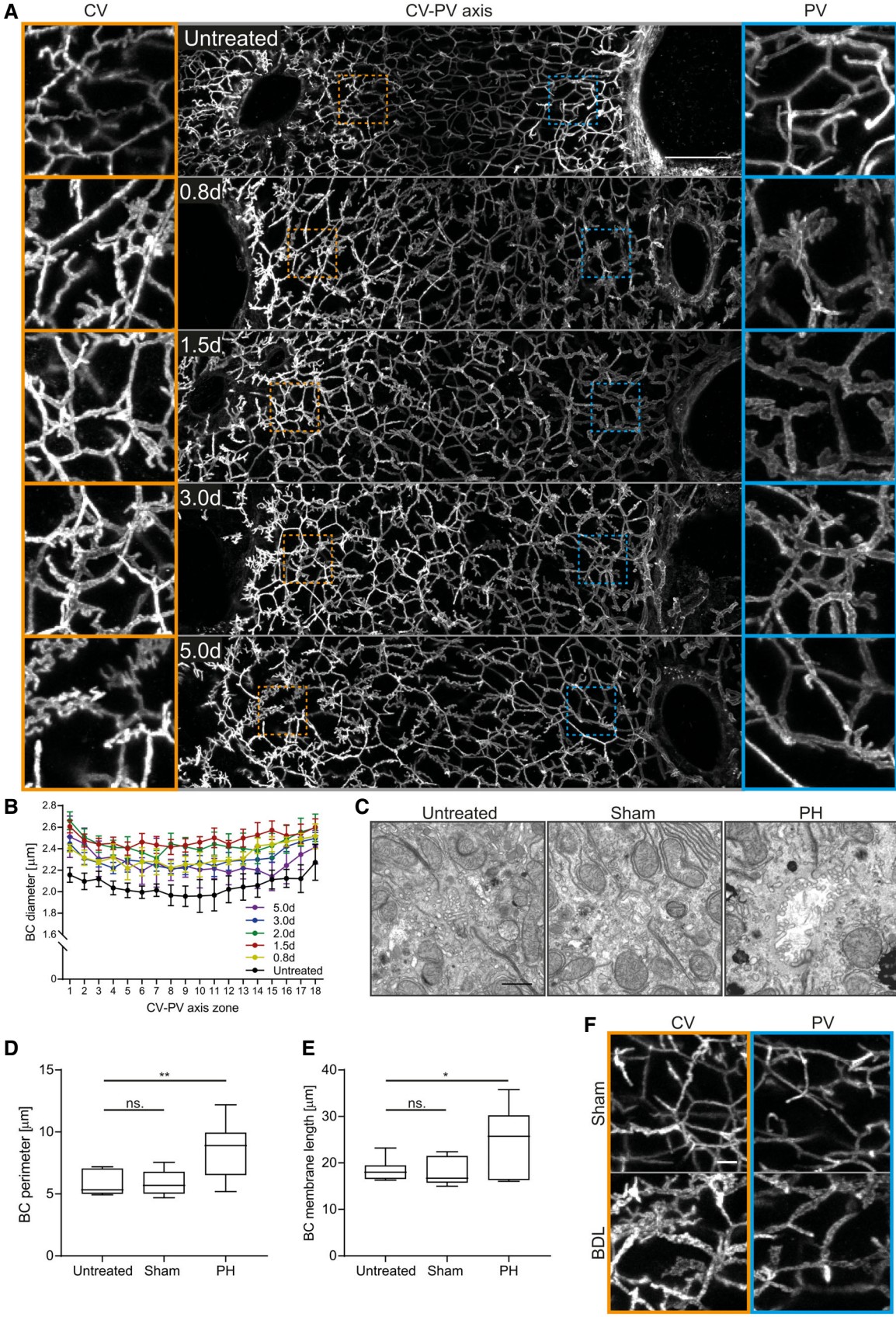

**Figure 1.**

**Figure 1.  The BC network transiently expands during liver regeneration.**

A      Fluorescence staining for the apical marker CD13 on liver tissue sections from untreated mice or animals at indicated time points post-PH. Shown are maximum projections of 50 μm z-stacks covering an entire CV-PV axis (CV, left; PV, right). Indicated regions of the CV (orange) and PV (blue) areas are shown as magnifications on the left and right, respectively.

B      Quantification of BC diameter within 18 zones along the CV-PV axis (zone 1, peri-central; zone 18, peri-portal) in livers from untreated animals or mice at indicated time points post-PH. The diameter was measured from 3D BC network reconstructions of IF image stacks of CD13 as shown in (A). The zones directly adjacent to the CV and PV were excluded from the analysis (~ 1 cell layer). Mean ± s.e.m, $n$ = 3–6 mice per time point. BC diameter of untreated mice vs. 0.8 day, $P$ = $1.62*10^{-11}$; untreated mice vs. 1.5 days, $P$ = $1.18*10^{-12}$; untreated mice vs. 2.0 days, $P$ = $5.69*10^{-11}$; untreated mice vs. 3.0 days, $P$ = $6.92*10^{-12}$; and untreated mice vs. 5.0 days, $P$ = $1.65*10^{-7}$.

C      EM images of BC on liver tissue sections of untreated mice (left) and 1.8 days after sham OP (middle) or PH (right).

D, E    Quantification of BC perimeter (D) and total BC membrane length (E) from EM images as representatively shown in (C). Box–whisker plot with median, 25–75 quartiles, and minimum/maximum error bars, $n$ = 5–6 mice per condition. In (D), BC perimeter of untreated vs. sham condition, $P$ = 0.94 (n.s.); untreated vs. PH condition, $P$ = 0.01. In (E), BC membrane length of untreated vs. sham condition, $P$ = 0.86 (n.s.); untreated vs. PH condition, $P$ = 0.03.

F      IF stainings of the apical marker CD13 on liver tissue sections from livers at 1 day after sham OP or BDL. Shown are maximum projections of 50 μm z-stacks in the CV (left, orange) and PV (right, blue) region. Images in (A) and (F) are background-subtracted.

Data information: Scale bars, 50 μm (A), 1 μm (C), and 5 μm (F).
Source data are available online for this figure.

pathway that responds to mechanical signals during liver regeneration. This choice is justified by the fact that YAP/TAZ depletion caused defects in liver regeneration (Lu *et al*, 2018). We hypothesize that YAP/TAZ could sense BA indirectly, through their effects on the apical actin cytoskeleton of hepatocytes, and drive the regenerative response (e.g., by control of proliferation (Lu *et al*, 2018), differentiation (Yimlamai *et al*, 2014; Fitamant *et al*, 2015), or ploidy (Zhang *et al*, 2017). To test whether YAP can sense BC network alterations, we examined its spatio-temporal dynamics using a specific antibody, validated by loss of signal upon YAP KO (Appendix Fig S3). We correlated YAP nuclear localization, as readout for its activity (Yagi *et al*, 1999; Zhao *et al*, 2007), with hepatocyte proliferation, detected by the cell cycle marker PCNA, as indicator for the onset of the cellular regenerative response. To account for previously reported spatial heterogeneities of proliferation within the liver lobule (Wu *et al*, 2011), we imaged the entire CV-PV axis at 17 time points (0.5–7 days) during regeneration (Fig 3A and B) and upon sham surgery (Appendix Fig S4A and B). Consistent with earlier reports (Yimlamai *et al*, 2014), YAP was highly expressed in cholangiocytes but at low levels in hepatocytes in the livers of untreated mice (Fig 3A) where it was primarily cytosolic, as expected for non-proliferating cells (LaQuaglia *et al*, 2016). During regeneration, however, YAP levels increased in hepatocytes as compared to untreated and sham-operated livers, as determined by both IF (Fig 3A, Appendix Fig S4A) and Western blot analysis (Appendix Fig S4C). The major fraction of YAP was still cytoplasmic, but, in addition, it was also detectable in nuclei of proliferating hepatocytes (Fig 3B, Appendix Fig S4B). Quantification of nuclear YAP and PCNA levels showed that nuclear YAP was specific to regenerating livers compared to sham-operated or untreated mice (Fig 3C). It started as an immediate response to liver resection, as early as 0.5 day post-PH, ceased with the proliferative wave at about 3–4 days (Fig 3C), and occurred throughout the entire CV-PV axis (Fig 3D).

To corroborate the morphological analysis, we used a subcellular fractionation approach. We isolated nuclear, membrane, and cytosolic fractions from liver tissue at 2 day after PH or sham surgery and determined the YAP content by immuno-blot analysis. In liver from sham-operated mice, the bulk of YAP was in the cytosol with low levels in the nuclear fraction (Fig 3F). In liver from PH, the fraction of nuclear YAP increased 2.5-fold (Fig 3F and G) as

compared to livers from sham-operated mice, consistent with the IF data of Fig 3.

The spatio-temporal dynamics of YAP activation remarkably correlate with those of BC network expansion (see Fig 1B). Interestingly, we found that YAP was enriched at the apical region of hepatocytes in regenerating liver (Fig 3E). The apical localization was detectable throughout the CV-PV axis, both in PCNA-positive and in PCNA-negative hepatocytes, and was specific to regenerating liver as compared to untreated and sham-operated mice (Appendix Fig S5A). Also, YAP displayed a particulate staining, suggesting that it may be spatially concentrated, e.g., on the actin cytoskeleton and/or organelles underneath the apical membrane (Fig 3E). The detection of YAP in the membrane fraction (Fig 3F and G) may reflect an association with organelles but also with the actin cortical mesh sedimented after high-speed centrifugation (see Methods and Protocols). Altogether, these results suggest that the apical localization and activation of YAP may be linked to the alterations of the BC network.

### YAP localizes to apical F-actin-enriched areas of hepatocytes during regeneration

To verify the apical localization of YAP, we visualized it by immunogold labeling and EM at 1.5 and 1.8 days post-PH (Fig 4A). Consistent with the IF, YAP was enriched at microvilli and in the sub-apical region of hepatocytes (Fig 4a′) as compared to the basolateral area (Fig 4a″). To quantify the enrichment, we determined YAP density at the apical and basolateral membrane of hepatocytes from EM images (6 × 6 images, ~ 30 × 30 μm; Appendix Fig S5B). The density of immunogold particles was on average 4.2-fold higher in the apical than the basolateral region (Appendix Fig S5C), demonstrating that YAP is enriched at the apical compartment during regeneration.

Yes-associated protein is a mechano-sensor that responds to alterations of the actin cytoskeleton (Dupont *et al*, 2011). From this perspective, its localization to the apical region of hepatocytes is ideal to sense alterations of biliary fluid dynamics and BC network expansion through changes of the actin cytoskeleton. As a pre-requisite, YAP should be enriched in areas of high F-actin content. To test for this, we used correlative light EM (CLEM) on liver tissue sections at 1.8 days post-PH (Fig 4B and C). F-actin was imaged at 186 nm resolution to identify F-actin-rich areas, whereas YAP was imaged at

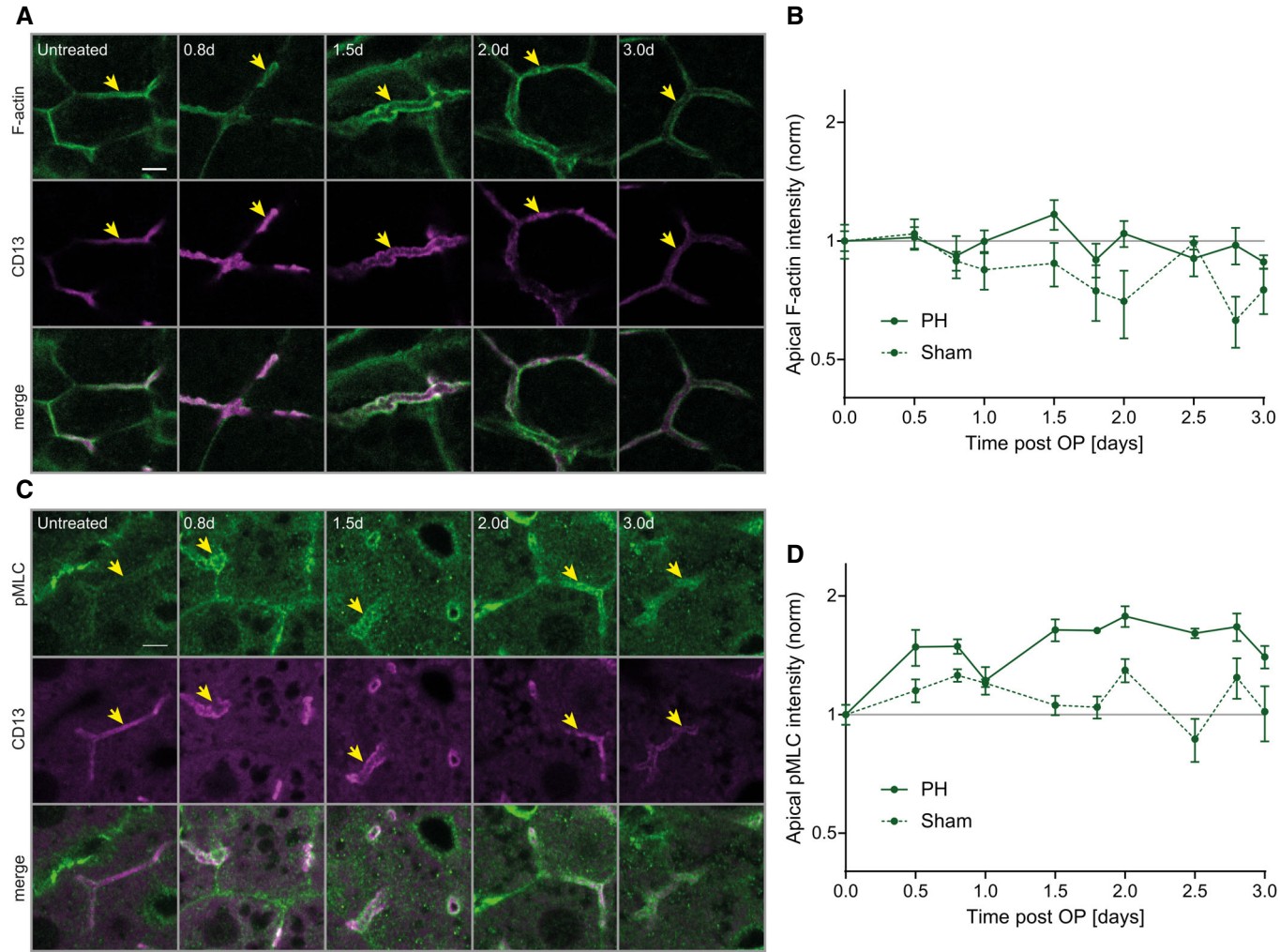

**Figure 2. BC network dilation is accompanied by increased apical myosin levels during regeneration.**

A–D Fluorescence stainings for F-actin (A) or pMLC (C) and the apical marker CD13 in the PV area on liver tissue sections from untreated mice or animals at indicated time points post-PH. Arrows indicate BC. Scale bar, 5 μm. Quantification of apical F-actin (B) and pMLC (D) intensity from images as representatively shown in (A and C) as well as Appendix Fig S2A and B, at indicated time points after PH (solid line) or sham OP (dashed line). Data are normalized to untreated animals (time point 0). Mean ± s.e.m, n = 3–5 (B) and n = 2–5 (D) mice per time point. Apical F-actin of sham vs. PH time course, P = 0.03; apical pMLC of sham vs. PH time course, P = 0.003.

Source data are available online for this figure.

1.1–2.6 nm resolution to visualize immunogold labeling. YAP was particularly enriched in the F-actin-dense sub-apical region (Fig 4c′ and c″, high F-actin levels; Fig 4c′″, low F-actin level, see also Appendix Fig S6). Gold particles were often associated with the microvilli in the BC, supporting the idea that YAP associates with the apical actin cytoskeleton. These results provide the first IF and EM detection of YAP at the apical compartment of hepatocytes *in vivo* and provide support to the idea that the actin-dependent mechano-sensory function of Hippo may link BA metabolism to growth control during liver regeneration.

To further verify that YAP interacts with the actin cytoskeleton and/or associated proteins, we immunoprecipitated YAP from liver lysate during regeneration (1.5 days post-PH, Appendix Fig S8) and analyzed the associated proteins by mass spectrometric analysis.

Unspecific interactions were determined by co-immunoprecipitation (co-IP) with rabbit IgG. A total of 82 proteins were consistently detected in two independent co-IPs (Appendix Table S1). KEGG pathway analysis revealed that most significantly enriched proteins were established YAP interactors of the Hippo pathway (Appendix Table S2), demonstrating the specificity of the approach. Further enrichment analysis for molecular functions using the DAVID database (Huang *et al*, 2009) revealed that actin-binding proteins were significantly enriched (*P*-value, $2.1 \times 10^{-9}$; Appendix Table S3). These include, for example, Trio and F-actin-binding protein, diaphanous and cingulin. We further found a large set of tight junction proteins as YAP interactors (e.g., Patj, Amotl1, Cgn, Magi1, Magi3, Mpp5, Mpdz, Tjp1, Tjp2, Tjp3) that may be involved in sensing mechanical or geometrical properties of BC. Our

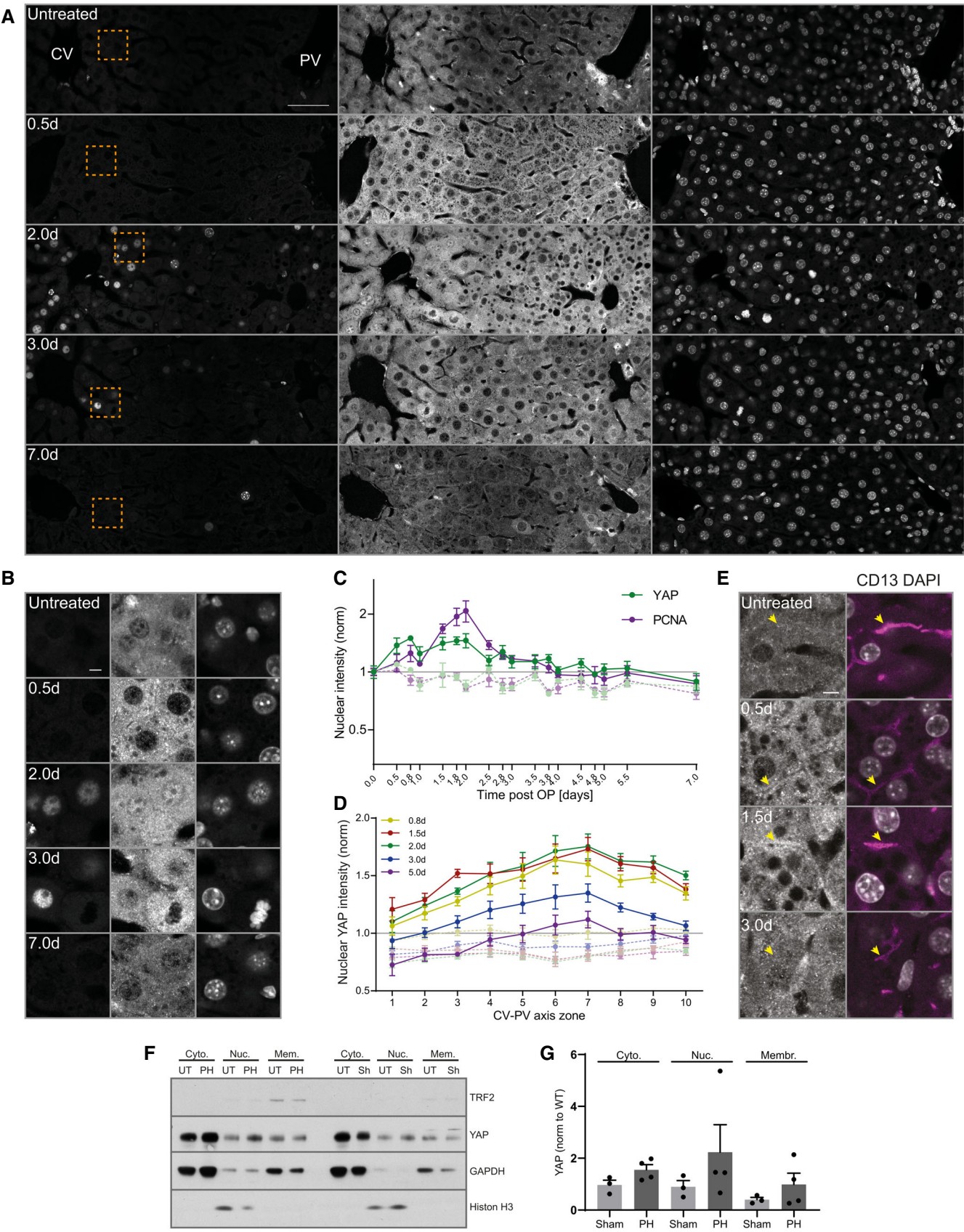

Figure 3.

**Figure 3. BC network expansion coincides with activation of YAP during regeneration.**

A, B Fluorescence stainings for YAP and PCNA and with the nuclear marker DAPI on liver tissue sections from untreated mice and animals at indicated time points post-PH. Images show an entire CV-PV axis (CV, left; PV, right). Indicated regions (orange squares) are shown as magnifications in (B). Note, the bright YAP fluorescence in the PV area stem from cholangiocytes of the bile duct.

C Quantification of the mean nuclear YAP (green) and PCNA (magenta) intensity from images of liver tissue sections at indicated time points after PH (solid line) and sham OP (dashed line) as representatively shown in (A) and Appendix Fig S4A. Data were normalized to untreated animals (time point 0). Mean $\pm$ s.e.m, $n = 3$–$5$ mice per time point. Nuclear YAP intensity of PH vs. sham time course, $P = 2.97*10^{-5}$. Nuclear PCNA intensity of PH vs. sham time course, $P = 7.51*10^{-6}$.

D Spatial analysis of the mean nuclear YAP intensity from images as representatively shown in (A) and Appendix Fig S4A, at indicated time points after PH (solid line) or sham OP (dashed line) in 10 zones within the CV-PV axis (zone 1, CV area; zone 10, PV area). Data are normalized to untreated animals (not shown). Mean $\pm$ s.e.m, $n = 3$–$5$ mice per time point. Nuclear YAP intensity of sham vs. PH mice at 0.8 day, $P = 5.35*10^{-5}$; 1.5 days, $P = 3.95*10^{-5}$; 2 days, $P = 5.99*10^{-5}$; 3 days, $P = 2.16*10^{-5}$; and 5 days, $P = 0.01$.

E Fluorescence stainings for YAP and CD13 and with the nuclear marker DAPI on liver tissue sections from an untreated mouse or animals at indicated time points post-PH. Arrows indicate BC. Note the enrichment of apical YAP at 0.5 and 1.5 days post-PH.

F Fractionation of liver tissue at 2 days after PH or sham (Sh) OP and untreated (UT) mice for nuclear (Nuc), cytoplasmic (Cyto), and membrane (Mem) fraction. TRF2, transferrin receptor 2 (membrane marker); histone H3 (nuclear marker); GAPDH, glyceraldehyde 3-phosphate dehydrogenase (cytoplasmic marker).

G Quantification of YAP in the nuclear, cytoplasmic, and membrane fraction from immunoblots as representatively shown in (f). YAP signal is normalized to the fraction marker (GAPDH, cytoplasm; histone H3, nuclei; and TRF2, membrane) and to the untreated condition (not shown). $n = 3$ (sham) or 4 (PH); data are shown as mean $\pm$ s.e.m. Differences between sham and PH conditions for cytoplasmic, nuclear, and membrane fractions are not significant.

Data information: Images in (A, B, and E) are background-subtracted. Scale bars, 50 μm (A) and 5 μm (B and E).
Source data are available online for this figure.

results showing that YAP is associated with components of the actin cytoskeleton support our hypothesis that it is regulated through the actin cytoskeleton during regeneration.

**Bile acid-induced activation of YAP is inhibited by perturbations of the actin cytoskeleton**

Given that YAP localizes to apical F-actin-rich areas in hepatocytes of regenerative liver and interacts with actin- and junction-associated proteins, we hypothesized that BA may activate YAP through modulation of the actin cytoskeleton, for example, through the acto-myosin system. We tested this idea on primary mouse hepatocytes. When cultured in collagen sandwich, these cells re-polarize forming BC *in vitro* (Zeigerer *et al*, 2017) and exhibit a cholestatic-like phenotype (Rippin *et al*, 2001), mimicking aspects of the metabolic state of the regenerative liver. Remarkably, YAP was not only cytoplasmic but also enriched at the apical region of the hepatocytes *in vitro* (Fig 5A, upper left), as observed during regeneration *in vivo* (see Figs 3E and 4). To recapitulate more closely the state of hepatocytes in the regenerating liver, we added the BA deoxycholic acid (DCA) at 200 μM (Fig 5A, lower left), a concentration that is comparable to serum levels in mice after PH (Naugler, 2014). DCA was sufficient to stimulate YAP nuclear translocation (1.7-fold increase as compared to DMSO control cells, Fig 5B), consistent with earlier observations (Anakk *et al*, 2013). DCA also induced a strong dilation of BC (Fig 5A, compare Control vs. DCA) and up-regulation of YAP (Appendix Fig S7A), as observed during regeneration *in vivo* (Appendix Fig S4C). Thus, the *in vitro* system recapitulates some of the properties of hepatocytes in the regenerating liver and, thus, proves suitable for studying YAP regulation.

We hypothesize that the stimulation of YAP nuclear translocation by DCA may depend on the actin cytoskeleton. Consistent with this, DCA increased pMLC levels 1.57-fold as compared to untreated cells (Fig 5C, Appendix Fig S7B). Incubation of hepatocytes with the Rho kinase inhibitor Y-27632 (Y27) potently reduced pMLC levels (Fig 5C, Appendix Fig S7B) and blocked the increase in nuclear YAP, reducing the effect of DCA by 47% (Fig 5A and B). These results suggest that the stimulatory effect of DCA requires the acto-myosin system, but the partial effect of Y27 argues that there may be an actin-independent component involved in BA-induced YAP

activation as well (e.g., by sensing osmotic pressure, membrane area/tension, cell–cell junctions).

Yes-associated protein is considered as an actin-dependent mechano-sensor (Dupont *et al*, 2011) that is sensitive to perturbations of the actin cytoskeleton, including actin nucleation, polymerization, and contractility (Dupont *et al*, 2011; Sansores-Garcia *et al*, 2011). The effects of Y27 on YAP activation suggest that YAP may also be regulated by the actin cytoskeleton in hepatocytes. To test whether YAP responds to specific alterations of the actin cytoskeleton, we screened a set of small molecule inhibitors targeting distinct actin properties, (i) acto-myosin contractility using the Rho kinase inhibitors Y27 and fasudil, (ii) actin nucleation and branching with the Arp2/3 inhibitor CK666 and the formin inhibitor SMIFH2, and (iii) F-actin polymerization using latrunculin A and cytochalasin D (Fig 5D). Although with different potency and activity, these pharmacological inhibitors showed a general trend for their effect on nuclear YAP levels in the hepatocytes *in vitro*. Cytochalasin D, latrunculin A, and SMIFH2 induced nuclear YAP translocation (Fig 5E), mimicking the effect of DCA, whereas the other drugs were either ineffective or inhibitory (Y27). The effects of cytochalasin D and latrunculin A in hepatocytes are in contrast to other previously reported systems (Dupont *et al*, 2011; Totaro *et al*, 2017). To verify the in/activation of YAP by the actin inhibitors, we inspected the expression of some target genes (Cyr61, CTGF, Ankrd1) by qPCR (Appendix Fig S7D). Consistently, Y27 and cytochalasin D have opposite effects, decreasing and increasing YAP target genes, respectively. However, latrunculin A reduced YAP target gene expression despite increasing nuclear YAP levels, indicating that nuclear translocation is not sufficient for target gene expression in this system. This is not too surprising as these genes are subjected to a much more complex transcriptional regulation that can be actin-dependent (MAL/SRF, Foster *et al*, 2017) and thus affected by the inhibitor.

Nevertheless, given that PH and DCA induce pMLC levels and that the latter is partially blocked by an acto-myosin inhibitor (Y27), we hypothesized that YAP may respond to alterations of the acto-myosin system, either directly or via a related morphological or physical BC property (apical membrane size, membrane tension, cell–cell junction integrity). In that case, YAP-activating compounds should also modify contractility. Quantification of pMLC levels of

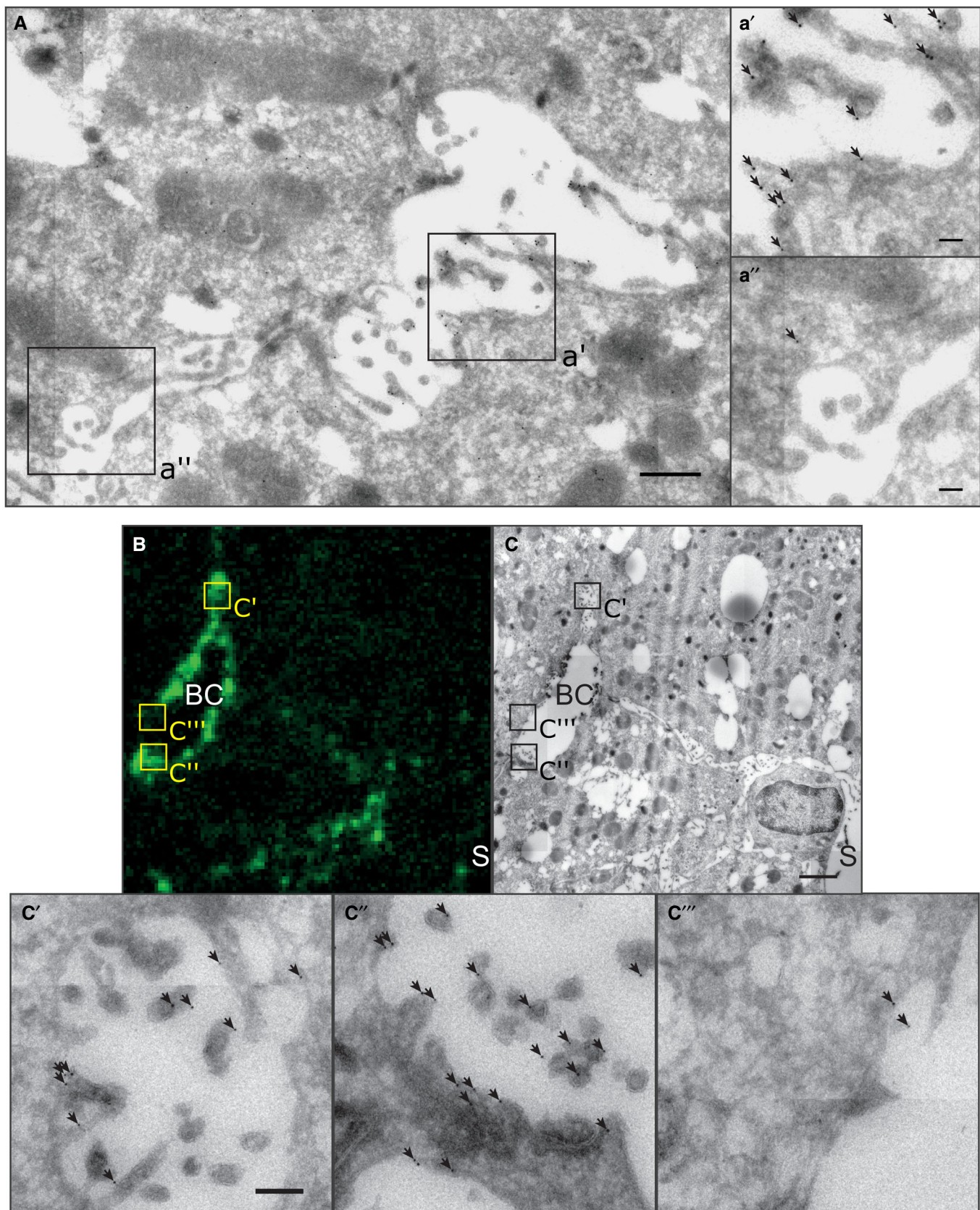

**Figure 4.**

**Figure 4. YAP enriches in F-actin rich areas in the apical compartment of hepatocytes during regeneration.**

A   EM image of immunogold-labeled YAP on liver tissue sections at 1.8 days post-PH. Image (A) shows a BC in between two hepatocytes. Indicated regions (black squares) of the apical (a′) and the basolateral (a″) membrane of the hepatocyte are shown as magnification. Arrows indicate gold particles.

B, C   Correlative light and electron microcopy of F-actin (B) and YAP (C) on a liver tissue section at 1.8 days post-PH. YAP was detected by immunogold labeling and F-actin by fluorescence staining with phalloidin. Indicated BC regions with high (c′, c″) or low (c‴) F-actin levels are shown as magnifications. Arrows indicate gold particles.

Data information: The image in (B) is background-subtracted. Scale bars, 0.5 μm (A), 0.1 μm (a′, a″), 2 μm (C), and 0.2 μm (c′). BC, bile canaliculus; S, sinusoid.

hepatocytes treated with the actin inhibitors revealed a general correlation between nuclear YAP localization and pMLC levels (Fig 5F, Appendix Fig S7C). Cytochalasin D and latrunculin A, which caused a pronounced activation of YAP (see Fig 5E), also produced the strongest increase in pMLC levels as compared to control cells (2.1–3.0-fold increase, Appendix Fig S7C). In contrast, both Y27 and Fasudil reduced pMLC levels in comparison with the control. However, the strength of pMLC reduction did not correlate with their effect on YAP. Even though the two inhibitors share the same target molecule, differential effects are not uncommon (Ichikawa *et al*, 2008). These results suggest that YAP may sense a BC property indirectly related to acto-myosin contractility.

Altogether, these *in vitro* results suggest that YAP nuclear translocation can be modified by perturbations of the actin cytoskeleton and pMLC levels. This is consistent with the hypothesis that YAP senses BA overload through changes of BC network structure that act, at least in part, via the actin-dependent regulation of the Hippo pathway.

**Fasudil-induced alterations of morphological/mechanical BC properties inhibit YAP activation during regeneration**

To validate our *in vitro* findings, we set to alter the actin cytoskeletal/morphological properties of BC during liver regeneration. We have previously shown that Fasudil efficiently inhibits BC contractility and bile flow *in vivo* (Meyer *et al*, 2017). Fasudil also proved to be a potent inhibitor of MLC phosphorylation in hepatocytes *in vitro* (see Appendix Fig S7C). Therefore, we applied fasudil by intraperitoneal injection to mice at 2 days after PH or sham operation for 1 h. First, we verified the effect of the Rock inhibitor on BC morphology by measuring their diameter (Fig 6A). Spatial analysis showed that fasudil strongly dilated the BC diameter in mice after PH (15.6%, zone 18) and moderately in the liver of sham OP mice (9.3%, zone 15; Fig 6B). This is consistent with the higher BC acto-myosin activity in the regenerating than control liver (Fig 2). Next, we tested the effect of fasudil on nuclear YAP levels during regeneration (Fig 6C). Fasudil reduced nuclear YAP levels during regeneration by up to 52% as compared to control mice (Fig 6D, red lines). Note that the sham operation alone caused a small increase in nuclear YAP levels that was also reduced by fasudil (Fig 6D, green lines). The results argue that YAP is indeed activated by BC properties in an actin-dependent manner during regeneration.

**A mechanistic model predicts a switch-like activation of YAP upon mechanical stimulation during regeneration**

Large and sustained increases in BA loads, as after PH, stimulate liver re-growth (Huang *et al*, 2006). In contrast, moderate fluctuations of BA concentrations, resulting, e.g., from circadian rhythm or

diet (Ma *et al*, 2009; Eggink *et al*, 2017), can be buffered by the metabolic activity of hepatocytes and do not result in YAP activation. This raises the question of how YAP can discriminate between physiological changes in BA loads and pressure in the BC vs. those that require liver growth.

Bile flow through BC is driven by both the osmotic pressure of actively pumped bile salts and osmolytes, and contractility of BC (Watanabe *et al*, 1991; Meyer *et al*, 2017). Upon PH without removal of the gall bladder, the total bile salt pool in the body is only marginally reduced, since intra-hepatic BA account for just 2–4% of the total pool (Setchell *et al*, 1997). Therefore, the liver remnant needs to transport the full bile salt pool through a reduced BC network, leading to an increased bile salt flux per liver weight (Vos *et al*, 1999) and, consequently, increased osmotic pressure, fluid inflow, increased fluid pressure, dilation of the apical membrane, and acto-myosin tension in the BC. It is currently technically impossible to measure pressure within the BC. However, it is possible to predict the changes in apical cortical tension and couple these to YAP nuclear translocation using a theoretical approach. Based on our experimentally measured spatial profile of BC diameter (see Fig 1B), we developed a biophysical model (see Supplementary Experimental Procedures) that predicts the apical cortical tension during regeneration from the osmotic pressure within the BC network. The model predicts a ~ 2-fold increase in apical tension throughout the entire CV-PV axis (Appendix Fig S9A) as an immediate response to liver resection within 0.8 day. Comparison of the inferred cortical tension and our experimentally measured apical pMLC levels shows a high correlation (Pearson correlation $r = 0.94$, Appendix Fig S9B), supporting the predictive power of the model.

Next, we coupled the biophysical model with a biochemical model of YAP regulation to predict nuclear YAP levels from cortical tension and strain. Based on reports on the regulation of YAP in the liver (Grijalva *et al*, 2014; Loforese *et al*, 2017; Lu *et al*, 2018), we considered five regulatory mechanisms: YAP synthesis, degradation, (in)activation, cytoplasmic sequestration, and nuclear–cytoplasmic shuttling (Fig 7A). Assuming simple mass action and Michaelis–Menten kinetics for the considered reactions (see Fig 7A), the model has 16 parameters. Of these, the compartment volumes of hepatocyte nuclei and cytoplasm were experimentally measured (Morales-Navarrete *et al*, 2015) (see Appendix Table S4). Five parameters were set to the value 1, as the analytical analysis of the model revealed that they do not affect the steady-state solution. The remaining nine parameters were estimated by fitting the quasi-steady-state solution of the model to 120 data points (nuclear and total YAP measurements; for details, see Supplementary Experimental Procedures and Appendix Table S4). To test the model, we predicted the nuclear and total YAP levels from the estimated apical cortical tension and compared the results to our experimental measurements (Appendix Fig S9C and D). Despite its relative simplicity, our

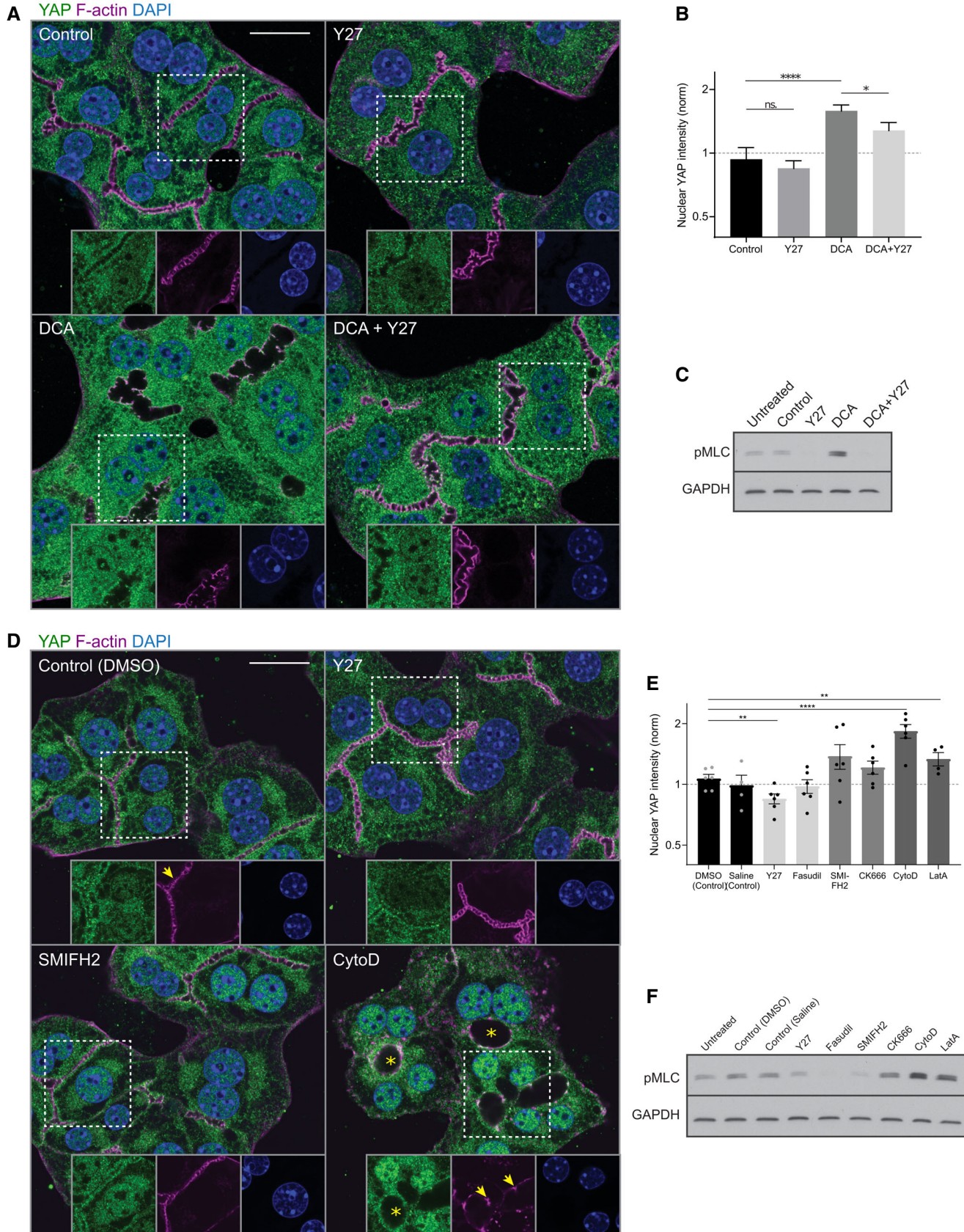

**Figure 5.**

**Figure 5. Bile acid-induced YAP activation is dependent on the acto-myosin system.**

A Fluorescence stainings for YAP (green) and F-actin (magenta) and with the nuclear marker DAPI (blue) of primary hepatocyte cultures treated with DMSO (control), Y27, deoxycholic acid (DCA), or DCA + Y27 for ~ 18 h. Indicated areas (dashed rectangle) are shown as magnifications in insets.

B Quantification of the mean nuclear YAP intensity from fluorescence images of primary hepatocytes treated with DMSO (control), Y27, DCA, or DCA + Y27 for ~ 18 h as representatively shown in (A). Data are normalized to untreated cells (not shown). Mean ± s.e.m., $n$ = 7; DCA vs. DMSO, $P$ = 8.98*10$^{-5}$ (****); DCA + Y27 vs. DMSO, $P$ = 0.05; Y27 vs. DMSO, $P$ = 0.55 (n.s.); DCA + Y27 vs. DCA, $P$ = 0.05 (*), $t$-test.

C Western blot of pMLC and GAPDH (loading control) of primary hepatocyte culture lysates. Cells were untreated or incubated for 18 h with the indicated compounds

D Fluorescence stainings of primary hepatocytes for YAP (green) and F-actin (magenta) and with the nuclear marker DAPI (blue). Cells were treated with DMSO (control), Y27, SMIFH2, or cytochalasin D (CytoD) for 6 h. Indicated areas (dashed rectangle) are shown as magnifications in insets. Note the dilation of canaliculi (asterisks) and fragmentation of F-actin (arrows) upon CytoD treatment.

E Quantification of the mean nuclear YAP intensity from images of primary hepatocytes treated with DMSO, saline, Y27, fasudil, SMIFH2, CK666, CytoD, or latrunculin A (LatA) for 6 h. Saline serves as control for fasudil, and DMSO serves as control for all other conditions. Inhibitors affecting similar actin processes are displayed in the same gray level. Saline, fasudil, CK666, and LatA conditions are not shown in (D). Data are normalized to untreated cells (not shown). Mean ± s.e.m., $n$ = 3–5; DMSO vs. Y27, $P$ = 0.002 (**); DMSO vs. CytoD, $P$ = 5.11*10$^{-7}$ (****); DMSO vs. LatA, $P$ = 0.01 (**); DMSO vs. CK666, $P$ = 0.15 (n.s.); DMSO vs. SMIFH2, $P$ = 0.11 (n.s.); saline vs. fasudil, $P$ = 0.92, $t$-test.

F Western blot of pMLC and GAPDH (loading control) in untreated or actin inhibitor-treated primary hepatocyte culture lysates. Inhibitor treatments are the same as in (E).

Data information: Scale bars, 20 μm (A and D).

mechanistic model reproduced the experimentally determined values of nuclear and total YAP levels over the time course of regeneration.

Finally, we used the model to infer the stimulus-response behavior, by analyzing YAP translocation to the nucleus as a function of BC cortical tension and membrane strain. The resulting stimulus-response curve (Fig 7B, black curve) is robust to changes of the kinetic rate constants. The nuclear YAP measured values match the theoretical curve very well (Fig 7B, colored symbols). The stimulus-response curve reveals a sharp threshold-like sigmoidal dependency where YAP translocates into the nucleus only if the relative mechanical stimulus exceeds ~ 1.75-fold (Fig 7B). This suggests that low and moderate fluctuations of BA load and cortical tension (e.g., diet-triggered) are tolerated and do not trigger YAP activation. In contrast, severe alterations of biliary pressure, such as after PH, robustly activate YAP in a switch-like manner to induce regeneration. Our model therefore predicts the existence of a threshold in the activation of YAP to ensure a switch from a resting to an activated state.

# Discussion

We described a novel mechanism whereby hepatocytes detect organ size during liver regeneration based on mechano-sensing of metabolic load through the apical actin cytoskeleton (Fig 7C). We demonstrate that liver resection induces structural and functional changes of the BC network, including expansion of the apical surface of hepatocytes and increase in acto-myosin contractility. These changes occur as a compensatory response to BA overload induced by tissue resection and are sensed by YAP, which is enriched at the apical domain of hepatocytes and translocates to the nucleus dependent on the integrity of the actin cytoskeleton. During regeneration, the restoration of the liver-to-body weight ratio also re-establishes the metabolic homeostasis, the structure of the BC network reverts to normal, and YAP returns to its physiological state (cytoplasmic localization). Thus, the BC network represents a self-regulatory mechano-sensory system that adapts to the overall metabolic demand of the body and acts as readout of tissue status.

The function of BA as regulators of liver regeneration has been demonstrated early on and mainly attributed to their signaling function via nuclear receptors (Huang et al, 2006). BA metabolism is a common target in clinical practice for treatment of a variety of human diseases to promote liver function. Whereas elevated BA levels promote hepatocyte proliferation, deprivation of BA results in a regenerative delay (Huang et al, 2006). How BA and YAP signaling are integrated to control liver regeneration remains an important question. Besides signal messengers, BA are also regulators of the Hippo pathway, acting via the induction of the scaffold protein IQGAP1 (Anakk et al, 2013). Our results implicate also the BC

**Figure 6. Inhibition of Rho kinase reduces YAP nuclear accumulation during regeneration.**

A Fluorescence stainings for the apical marker CD13 on liver tissue sections from mice at 2 days after sham OP or PH, treated with saline (control) or the Rho kinase inhibitor fasudil for 1 h. Shown are maximum projections of 50-μm stacks in the PV area.

B Quantification of BC diameter within 18 zones along the CV-PV axis (zone 1, peri-central; zone 18, peri-portal) from mice at 2 days after sham OP (green) or PH (red), treated with saline (control, solid line) or fasudil (dashed line). Diameter was measured from 3D BC network reconstructions of image stacks as representatively shown in (A). The zones directly adjacent to the CV and PV were excluded from the analysis (~ 1 cell layer). Mean ± s.e.m, $n$ = 7–8 mice per condition from three independent experiments. BC diameter of sham-operated mice treated with saline vs. fasudil, $P$ = 6.59*10$^{-8}$; BC diameter of PH mice treated with saline vs. fasudil, $P$ = 2.31*10$^{-14}$.

C Fluorescence stainings for YAP and PCNA and with the nuclear marker DAPI on liver tissue sections from untreated mice or animals at 2 days after sham OP or PH, treated with saline (control) or fasudil for 1 h. Indicated regions (dashed rectangle) are shown as magnifications in insets.

D Quantification of the mean nuclear YAP intensity within 10 zones along the CV-PV axis, (zone 1, peri-central; zone 10, peri-portal) from IF images as representatively shown in (C). Data were normalized to untreated animals. Mean ± s.e.m, $n$ = 7–8 mice per condition from three independent experiments. Nuclear YAP intensity of sham-operated mice treated with saline vs. fasudil, $P$ = 0.13 (n.s.); nuclear YAP intensity of PH mice treated with saline vs. fasudil, $P$ = 0.0004.

Data information: Scale bars, 5 μm (A) and 50 μm (C).
Source data are available online for this figure.

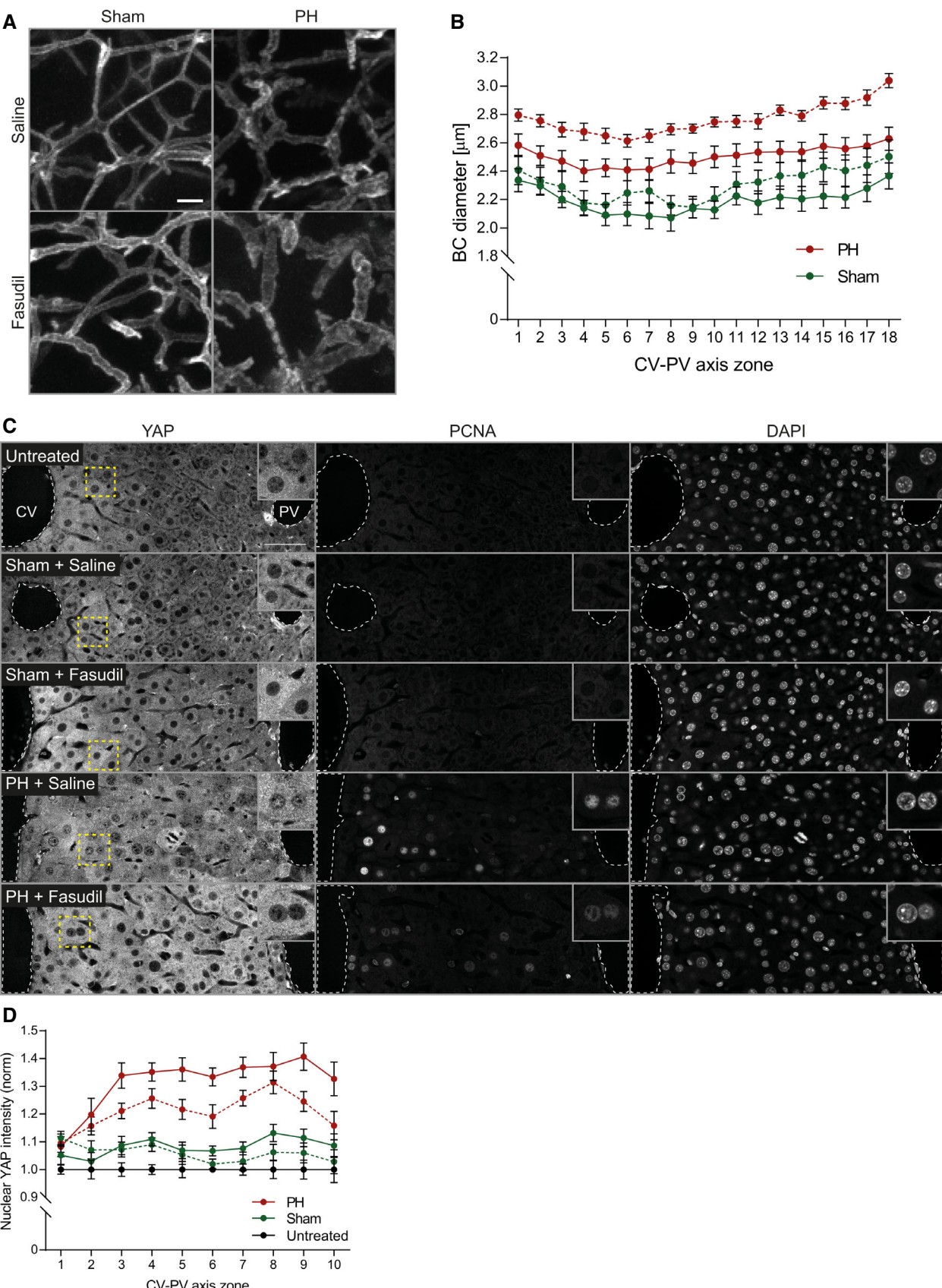

**Figure 6.**

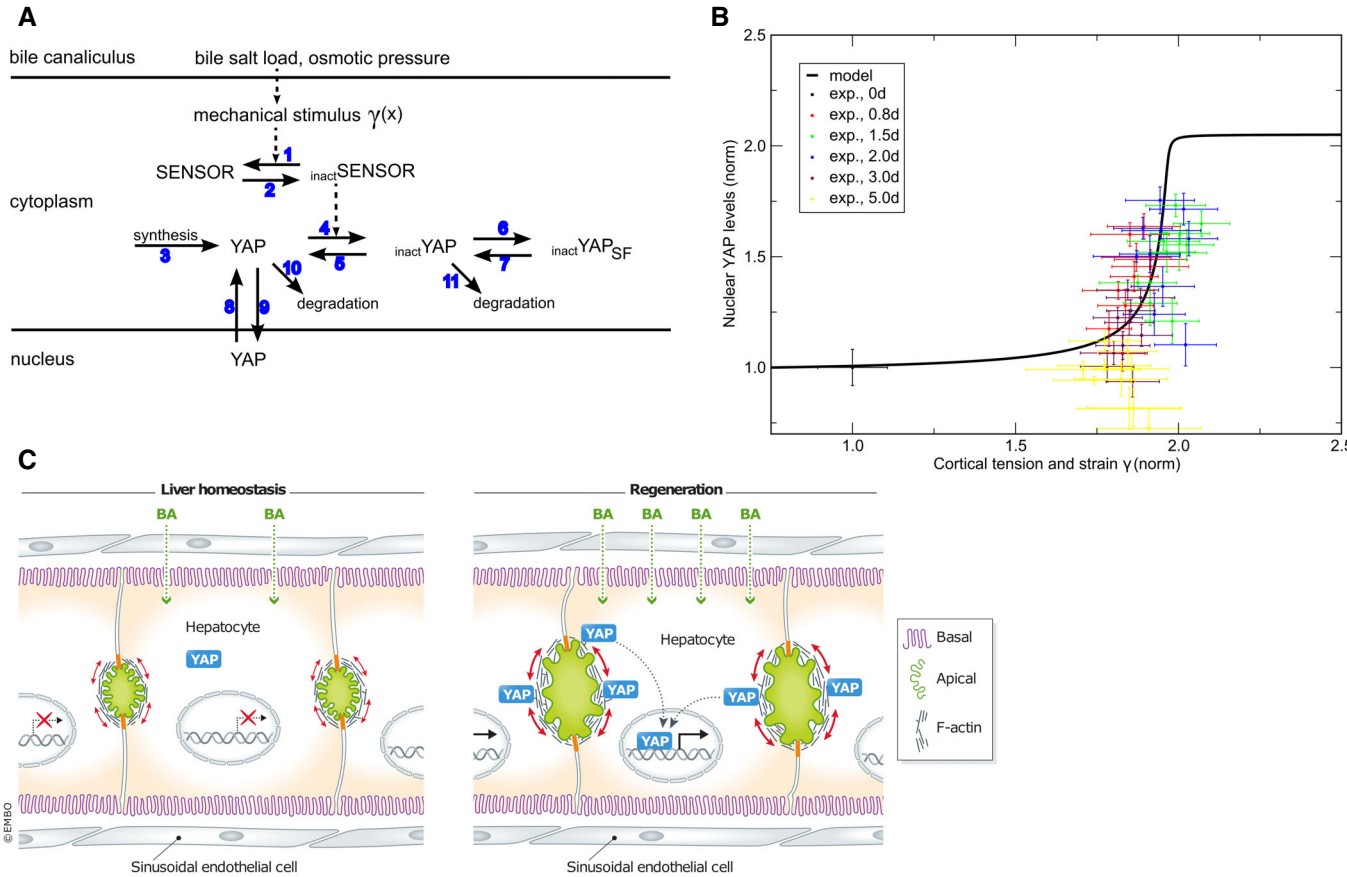

**Figure 7. Model of mechano-sensing of metabolic status during liver regeneration.**

A   Model of YAP regulation by mechanical stimulation through expansion of the BC. Arrows denote reactions or transport steps. Blue numbers label individual processes and associated parameters as listed in Appendix Table S4. The pathway combines a signaling cascade with regulated cytoplasmic retention and is here illustrated based on inactivation by a mechano-sensor, SENSOR; $_{inact}$SENSOR, inactivated mechano-sensor; $_{inact}$YAP, inactivated YAP; SF, sequestration factor; but alternative molecular states are possible. See Supplementary Experimental Procedures for model equations and analysis.

B   Model prediction (solid black line) of cortical tension and strain and nuclear YAP levels reveals a sigmoidal stimulus–response curve. Experimental data (symbols, shown as mean ± s.e.m.) of nuclear YAP are reproduced from Fig 3D and mapped to cortical tension and strain levels using experimental data from Figs 1B and 2D. For details see legend of Figs 1B, 2D, and 3D.

C   Schematic drawing of YAP regulation by BA through canalicular cortical tension and strain. Tissue resection by partial hepatectomy induces a BA overload which elevates osmotic pressure and cortical tension of bile canaliculi. Concomitant changes of apical acto-myosin properties recruit YAP to the apical cortex where it is activated for nuclear translocation.

network in such a mechanism, by playing a mechano-sensory role through the actin cytoskeleton, linking BA metabolism, liver tissue structure, and Hippo signaling. The precise molecular mechanisms underlying the mechano-sensing function of the actin cytoskeleton which is read out by YAP is unclear in this system. The finding that inhibition of acto-myosin activity by Fasudil is sufficient to inhibit YAP nuclear localization during liver regeneration supports a regulation of YAP by the acto-myosin system. However, it does not exclude other mechanisms and pathways that could sense apical membrane size, membrane tension, cell–cell junction integrity, and, in general, the actin cytoskeleton. For example, the MAL/SRF transcription factors are also suitable candidates since they respond to actin dynamics and regulate YAP/TAZ target genes (Foster *et al*, 2017). In addition, it remains unclear if nuclear receptors also signal through or affect the actin cytoskeleton and act in concert with the suggested acto-myosin-mediated signaling branch.

A remarkable feature of liver tissue organization is that all hepatocytes are connected via the BC, bridging the sub-cellular to the tissue level. Such organization permits the communication of global tissue properties, e.g., metabolic status, to each individual cell within the lobule, to control cell behavior collectively. Key for the signaling function of the BC network is the dynamic nature of the apical plasma membrane and its associated actin cytoskeleton of hepatocytes, which can promptly respond to metabolic changes. The rate of water flow into BC is subjected to a complex regulation in response to BA transport but also aquaporin-mediated permeability, leading to changes in osmotic pressure and bile flow (Marinelli *et al*, 2019). Bile pressure forms a gradient within the lobule varying ∼ 30-fold from the CV to the PV area (Meyer *et al*, 2017). Such variation needs to be balanced by the tension of the acto-myosin system. This implies that changes of pressure need to

be sensed, evoking the need for a mechano-sensory system. The same system may conceivably operate during liver regeneration.

Our mathematical model predicts a sharp pressure-dependent threshold in the activation of YAP to ensure a robust switch from an inactive to an active state in response to the metabolic overload. This means that fluctuations of BA concentrations in the physiological range could be compensated by the metabolic activity of hepatocytes, whereas large and sustained changes, as after PH, trigger the Hippo pathway in a switch-like manner to induce regeneration. Our mechanistic mathematical model of signal transduction through the Hippo pathway triggered by a mechanical stimulus may be applicable to morphogenesis more generally, e.g., growth of the fly wing imaginal disk in response to apical membrane tension (Pan *et al*, 2016).

Several signaling pathways are activated during liver regeneration, and the Hippo pathway is only one of these. Its absolute requirement in liver regeneration is unclear (Lu *et al*, 2018), but our experimental observations and mathematical model could in principle be applied to other signaling pathways that can be activated in response to mechanical stimuli, as discussed above. The precise function of YAP and how its activity is controlled on the molecular level are important future questions. In the homeostatic liver, YAP has pleiotropic functions that go beyond its classic role as regulator of proliferation. YAP function has been implicated in hepatocyte differentiation (Yimlamai *et al*, 2014; Fitamant *et al*, 2015), ploidy (Zhang *et al*, 2017), identity/metabolic zonation (Fitamant *et al*, 2015), and cell size (Tumaneng *et al*, 2012). A more detailed analysis of liver tissue function and architecture in YAP KO mice after PH may link our findings to its downstream functions paving the way toward a systems understanding of liver regeneration. A wide array of apical actin interacting, polarity, and cell junction-associated proteins have already been reported to regulate the Hippo pathway (Chen *et al*, 2010; Oka *et al*, 2010). Given the collective properties of the apical plasma membrane and associated actin cytoskeleton, a systems-level analysis of its complex super-molecular network will be required to identify the precise underlying molecular mechanisms. Our reported YAP interactors, including actin and cell–cell junction-associated proteins, provides a list of candidate molecular players potentially involved.

# Materials and Methods

## Reagents and Tools table

| Reagent/resource | Reference or source | Identifier or catalog number |
|---|---|---|
| **Experimental models** | | |
| C57BL/6JOlaHsd mice | Harlan Laboratories | |
| YAP^fl/fl mice | Zhang *et al* (2010) | |
| **Antibodies** | | |
| CD13 (rat, monoclonal, dilution for IF 1:400) | Acris Antibodies | SM2298P |
| PCNA (mouse monoclonal, Alexa Fluor 488 conjugate, dilution for IF 1:200) | Cell Signaling Technologies | 8580 |
| YAP (rabbit polyclonal, dilution for IF 1:2,000 and for EM 1:50) | Laboratory of Prof. Elly Tanaka, IMP, Vienna, Austria | |
| pMLC (rabbit, polyclonal, dilution for IF 1:100) | Abcam | ab2480 |
| YAP (rabbit polyclonal, dilution for WB 1:1,000) | Cell Signaling Technology, USA | 4912 |
| GAPDH (mouse, monoclonal, dilution for WB 1:2,000) | Sigma-Aldrich | G8795 |
| Alpha-tubulin (mouse, monoclonal, dilution for WB 1:1,000) | Sigma-Aldrich | T6199 |
| Histone H3 (rabbit polyclonal, dilution for WB 1:1,000) | Millipore | 06-755 |
| Transferrin receptor 2 (rabbit polyclonal, dilution for WB 1:1,000) | Abcam | 80194 |
| **Oligonucleotides and other sequence-based reagents** | | |
| Cyr61_F (CTGGCATCTCCACACGAGTTAC) | Diepenbruck *et al*, Journal of Cell Science, 2014 | |
| Cyr61_R (TGCCCTTTTTTAGGCTGCTG) | Diepenbruck *et al*, Journal of Cell Science, 2014 | |
| CTGF_F (CACAGAGTGGAGCGCCTGTTC) | Andrianifahanana *et al*, FASEB Journal, 2016 | |
| CTGF_R (GATGCACTTTTTGCCCTTCTTAATG) | Andrianifahanana *et al*, FASEB Journal, 2016 | |
| Ankrd1_F (GGAACAACGGAAAAGCGAGAA) | PrimerBank database (Wang *et al*, Nucleic Acids Research, 2012) | Primer bank ID 133893293c1 |
| Ankrd1_R (GAAACCTCGGCACATCCACA) | PrimerBank database (Wang *et al*, Nucleic Acids Research, 2012) | Primer bank ID 133893293c1 |

**Reagents and Tools table** (continued)

| Reagent/resource | Reference or source | Identifier or catalog number |
| --- | --- | --- |
| GAPDH_F (CACTGAGCATCTCCCTCACA) | Zeigerer *et al*, Cell Reports, 2015 | |
| GAPDH_R (GTGGGTGCAGCGAACTTTAT) | Zeigerer *et al*, Cell Reports, 2015 | |
| **Software** | | |
| MotionTracking software | Morales-Navarrete *et al* (2015) (http://motiontracking.mpi-cbg.de) | |
| IMOD | Kremer *et al*, Journal of Structural Biology, 1996 (http://bio3d.colorado.edu/imod) | |
| GraphPad | GraphPad Software, Inc. | |
| Fiji | Schindelin *et al* (2012) | |
| Copasi | Hoops *et al*, Bioinformatics, 2006 | |
| Morpheus | Starruß *et al*, Bioinformatics, 2014 | |
| MASCOT software | Matrix Science, London, UK | |
| Scaffold software | Proteome Software Inc., Portland, US | |

## Methods and Protocols

### Animal work

Animal experiments were performed on 8–12-week-old, male, C57BL/6JOlaHsd (Harlan laboratories) mice at the MPI-CBG (Dresden, Germany). PH and BDL experiments were performed between ~ 8 am and 1 pm. Experiments were conducted in accordance with German animal welfare legislation and in strict pathogen-free conditions in the animal facility of the MPI-CBG. Protocols were approved by the Institutional Animal Welfare Officer (Tierschutzbeauftragter), and all necessary licenses were obtained from the regional Ethical Commission for Animal Experimentation of Dresden, Germany (Tierversuchskommission, Landesdirektion Dresden). Mice were fasted 6 h prior to sacrifice (water *ad libitum*).

### PH

PH was performed based on Mitchell and Willenbring (2014) but without removal of the gall bladder (left and right median lobes resected individually). Sham-operated mice received the same treatment but without liver resection.

### YAP knockout

A conditional hepatocyte-specific mosaic YAP knockout was induced by adeno-associated virus-mediated expression of iCre recombinase using AAV/DJ-pALB(1.9)-iCre (Vector Biolabs, USA) in YAP$^{fl/fl}$ mice (Zhang *et al*, 2010). AAV/DJ-pALB(1.9)-eGFP (Vector Biolabs, USA) served as control.

### Fasudil administration

Mice received 30 mg/kg fasudil or saline intravenously and were sacrificed 1 h after.

### Bile duct ligation

Bile duct ligation was performed as described in Tag *et al* (2015) and sacrificed 1 day post-surgery. Sham-operated mice received the same treatment without BDL.

### Tissue fixation

Anaesthetized mice were perfused transcardially with 4% PFA/ 0.1% Tween-20/PBS. Tissue was post-fixed in 4% PFA/0.1% Tween-20/PBS for ~ 48 h at 4°C for IF or in 1% glutaraldehyde/3% PFA/PBS for 1.5 h for morphological EM.

### Primary hepatocytes

Hepatocytes were isolated and cultured as described in Godoy *et al* (2013) and Zeigerer *et al* (2017), respectively. Cells were treated with 20 μM Y27, fasudil, SMIFH2, CK666, or 5 μM latrunculin A or cytochalasin D for 6 h. For BA experiments, cells were pre-incubated with 20 μM Y27 or DMSO (control) for 1 h, followed by 16–18 h incubation with DMSO, 200 μM DCA, 20 μM Y27, or 20 μM + 200 μM DCA. Cells were lysed for biochemical analysis or fixed with 4% PFA for 30 min.

### IF stainings

One hundred micrometer liver slices were permeabilized with 0.5% Triton X-100 for 1 h, quenched with 10 mM $NH_4Cl$ for 30 min, blocked with 0.2% gelatin/300 mM NaCl/0.3% TX100/ PBS, and incubated with primary (see Reagents and Tools table) and fluorescently conjugated secondary antibody in blocking buffer for 48 h each. Co-staining of CD13 with PCNA and YAP or pMLC required additional antigen retrieval in citric acid or EDTA buffer at 80°C for 1 h, respectively. To preserve CD13 staining during retrieval, tissue was pre-fixed with 4% PFA for 20 min. Following retrieval, sections were stained with primary antibodies as described above. Sections were mounted in 90% glycerol (2D microscopy) or cleared with SeeDB (3D imaging) as previously described (Ke *et al*, 2013). IF staining of hepatocytes was performed as described previously (Zeigerer *et al*, 2017). Antibody specifications and dilutions are listed in the Reagents and Tools Table.

### Microscopy

Samples were imaged on a Zeiss LSM780 confocal microscope as 2 × 1 image tiles covering an entire CV-PV axis as single-plane

image or 30 μm z-stacks (voxel size of $0.28 \times 0.28 \times 0.3$ μm) as described in (Meyer et al, 2017; Morales-Navarrete et al, 2015).

### RNA extraction and RT–PCR

RNA was extracted from primary hepatocyte sandwich cultures using TRIzol (Invitrogen) and purified with the RNeasy Kit (Qiagen). RNA was DNase I-treated and reverse-transcribed using SuperScript III Reverse Transcriptase (Invitrogen) and random hexamer primers. The quantitative PCR was performed using SYBR Green (Thermo Scientific) and the primers listed in the Reagents and Resource Table. Reactions were run on a Roche LightCycler with the following thermal cycling condition: 95°C for 15 min, 40 cycles of 95°C for 15 s, 60°C for 15 s, and 72°C for 15 s. The quantification cycle ($C_q$) was extracted, and the relative expression of each gene (Cyr61, CTGF, Ankrd1) was calculated using the comparative $C_q$ method (normalized to GAPDH).

### Liver tissue fractionation

The protocol was adapted from Cox and Emili (2006). Liver tissue from untreated, sham-operated, or regenerating livers (2 days post-OP) was homogenized in 1 ml 250 mM sucrose buffer [0.25 M sucrose, 50 mM Tris–HCl (pH 7.4), 5 mM $MgCl_2$, 1 mM DTT, 25 μg/ml spermine, 25 μg/ml spermidine, protease inhibitors] and centrifuged at 1,000 $g$ for 15 min. The pellet (nuclear fraction) was resuspended in 1 ml of 250 mM sucrose buffer, centrifuged at 1,000 $g$ for 15 min, washed twice with 1 ml 250 mM sucrose buffer, and centrifuged at 1,000 $g$ for 15 min. The supernatant from the initial tissue homogenate was centrifuged for 1 h at 100,000 $g$ to yield the cytosol (supernatant) and membrane (pellet) fractions. The membrane pellet was resuspended in lysis buffer (20 mM Tris–HCl pH 7.5, 150 mM NaCl, 1 mM EDTA, 1 mM EGTA, 1% SDS, 1% NP-40, 1 mM DTT, protease inhibitors) for 1 h at 4°C, centrifuged at 9,000 $g$ for 30 min to remove insoluble material.

### YAP co-immunoprecipitation

Liver tissue from a mouse at 1.5 days after PH or sham OP was mechanically homogenized in IP buffer (20 mM Tris–HCl pH 7.5, 100 mM NaCl, 1 mM EDTA, 0.5% Triton X-100, protease inhibitors, 5 μM latrunculin A) and incubated at 4°C for 30 min on a rotator. Cell debris was removed by centrifugation at 10,000 $g$ for 10 min at 4°C. The supernatant was centrifuged at 4°C for 45 min at 150,000 $g$ and the protein concentration of all conditions adjusted. Rabbit YAP antibody (Elly Tanaka, IMP Vienna) and rabbit IgG control antibody were covalently cross-linked to Protein G Dynabeads (Life Technologies) using bis(sulfosuccinimidyl)suberate (BS[3]) reagent (Thermo Scientific) according to manufacturer's instructions. The antibody-coupled beads were added to the liver lysate supernatant and incubated at 4°C for 45 min on a rotator. The beads were washed 4 times with 1 ml ice-cold IP buffer, resuspended in sample buffer (50 mM Tris–HCl pH 6.8, 10% glycerol, 2% SDS, 0.05% bromphenol blue), and incubated at 90°C for 5 min. The supernatant was denatured by addition of DTT to a final concentration of 100 mM and incubation at 90°C for 10 min.

### Mass spectrometry

Protein eluates from co-IP experiments were separated by SDS–PAGE and Coomassie-stained, and each gel lane was cut into 10 slabs, each of which was in-gel-digested with trypsin. GeLC analysis

was performed on a nano-HPLC UltiMate 3000 interfaced online to a LTQ Orbitrap Velos hybrid tandem mass spectrometer (both Thermo Fisher Scientific, Germany). Database search was performed against mouse entries in UniProt DB using MASCOT software (Matrix Science, London, UK) under the following settings: enzyme specificity—trypsin; two miscleavages; mass tolerance 5 ppm and 0.5 Da for precursor and fragments, respectively; and variable modifications—oxidation of methionine, acetyl of the protein N-terminus, and propionamide of cysteine. Scaffold software (Proteome Software Inc., Portland, US) was used to validate MS/MS-based protein identifications. Peptide identifications were accepted if they could be established at > 95.0% probability—protein identifications at greater than 99.0%—and contained at least two identified peptides. Label-free quantification of relative abundance of proteins was performed for Scaffold-validated entries using the MaxQuant software. In total, data from two independent co-IP experiments were analyzed. Unspecific proteins detected in IgG control conditions were removed from the protein list. The abundance of proteins identified was normalized to an internal standard GluFib as well as the bait YAP.

### Western blot

SDS–PAGE-separated liver lysates were transferred onto nitrocellulose membrane. Proteins were immuno-detected using chemiluminescence. The relative density of bands was quantified using the Fiji software. For quantification of YAP protein in liver tissue fractions, YAP signal was normalized to the fraction marker (GAPDH, histone H3, TRF2) and each condition (sham, PH) was normalized to the untreated control. Antibody specifications and dilutions are listed in the Reagents and Tools Table.

### EM

For morphological EM, 70 nm liver sections were imaged as $10 \times 10$ grids (image size $19 \times 19$ μm, pixel size 9 nm) on a Philips Tecnai 12 EM (FEI, USA). For immuno-EM, sections were immuno-labeled using YAP antibody (Reagents and Tools table) and goat anti-rabbit 10 nm gold-coupled IgG as previously described (Paridaen et al, 2015) and imaged at 1.1–2.6 nm resolution. For CLEM, sections were additionally incubated with Alexa Fluor 488-conjugated phalloidin and imaged at 0.186 μm/pixel resolution on a Zeiss Axioplan 2 light microscope.

### Image analysis and mathematical modeling

See Appendix.

# Data availability

The source code for the computational model of YAP activation is provided as Code EV1.

**Expanded View** for this article is available online.

## Acknowledgements

We thank the Biomedical Services (Jussi Helppi, Anne-Muench Wuttke, and Barbara Langen), Light Microscopy Facility (Jan Peychl), EM Facility (Jean-Marc Verbavatz) of the MPI-CBG, and the Centre for Information Services and High Performance Computing (ZIH) of the TU Dresden for the

generous provision of computing power. This work was financially supported by the Virtual Liver (http://www.virtual-liver.de, grant #315757), Liver Systems Medicine (LiSyM, grant #031L0038), DYNAFLOW (grant #031L008A to L.B. and M.Z.) initiatives, funded by the German Federal Ministry of Research and Education (BMBF), the BMBF grant on "a systems microscopy approach to tissues and organ formation" (grant #031L0044), the European Research Council (grant #695646), and the Max Planck Society (MPG).

## Author contributions

KM and MZ conceived the project, and KM designed most of the experimental strategy. KM, with help of SS, conducted the experiments. MW-B performed immuno-EM and CLEM. UD trained KM in PH. HM-N and YK developed the image analysis algorithms. EMT provided the YAP antibody, a key reagent for the study. LB and YK developed and analyzed the mathematical model. KM and MZ wrote the manuscript.

## Conflict of interest

The authors declare that they have no conflict of interest.

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
