## [Review Process File · Molecular Systems Biology]

Bile canaliculi remodeling activates YAP via the actin cytoskeleton during liver regeneration

Kirstin Meyer, Hernan Morales-Navarrete, Sarah Seifert, Michaela Wilsch-Braeuninger, Uta Dahmen, Elly Tanaka, Lutz Brusch, Yannis Kalaidzidis and Marino Zerial

Review timeline:	Submission date:	3rd May 19
	Editorial Decision:	11th Jun 19
	Revision received:	3rd Dec 19
	Editorial Decision:	20th Dec 19
	Revision received:	19th Jan 20
	Accepted:	23rd Jan 20

Editor: Maria Polychronidou

Transaction Report:

1st Editorial Decision

11th Jun 19

Thank you again for submitting your work to Molecular Systems Biology. We have now heard back from the three referees who agreed to evaluate your study. As you will see below, the reviewers acknowledge that the presented findings seem potentially interesting. They raise however a series of concerns, which we would ask you to address in a major revision.

I think that the reviewers' recommendations are rather clear and there is therefore no need to repeat the comments listed below. Importantly, reviewers #1 and #3 point out that additional analyses are required to better support the main conclusions and they provide constructive suggestions in that regard.

All issues raised by the reviewers need to be satisfactorily addressed. As you may already know, our editorial policy allows in principle a single round of major revision and it is therefore essential to provide responses to the reviewers' comments that are as complete as possible. Please feel free to contact me in case you would like to discuss in further detail any of the issues raised by the reviewers.

REFeree REPORTS

Reviewer #1:

The manuscript by Meyer and collaborators addresses the topic of mechanosensing in liver by adopting quite a unique and original approach.

The experimental design is extremely neat and clear, the results are very interesting and potentially groundbreaking.

The conclusions, though, are not always well supported by the results shown.

The authors rely mostly on high resolution microscopy, where YAP visualisation is suboptimal. Biochemical experiments would highly increase the impact of the manuscript.

Briefly, the authors show bile canaliculi (BC) undergo dilation and branching right after hepatectomy and reverse at 3.5 days PH.

By high resolution microscopy, they measure the changes in BC perimeter and total membrane length, they show both increased to a similar extent and conclude that the phenomenon is due to an increase in total apical membrane of hepatocytes.

They hypothesise that this is a response to increased BA levels to induce liver growth during regeneration and found it is associated with a concomitant increase in apical acto-myosin contractility.

Since YAP is downstream of acto-myosin stress, the authors proceed to demonstrate that BA accumulation causes an activation of YAP in hepatocytes, thus causing their proliferation during regeneration.

The main result, which could be potentially groundbreaking, is that YAP localizes to the apical membrane in actin-rich fibers. In fact, so far no direct evidence of YAP being in direct contact with the cytoskeleton was given *in vivo*. This result probably deserves some deeper characterization. Also, a closer understanding of how this process is correlated with YAP nuclear translocation is probably needed. This is not clear at all.

Then the authors move to *in vitro* primary hepatocytes culture to show bile acid activates YAP through acto-myosin axis. This result was already proven in tumor cells (doi:10.1126/science.aav0173).

Finally, the manuscript proposes a model of YAP activation during liver regeneration.

In this reviewer's opinion, the manuscript is definitely of interest for the scientific community, but additional experiments would be needed to strengthen the results.

Major concerns:

- 1) Figs. 1d, e lack the statistical analysis. This way it is not clear how significant the changes are.
- 2) Results in figure 2 are not so striking. I am not sure this is enough to say there is an increased actomyosin contractility.
- 3) In figure 3 and S4, the authors claim YAP translocation to the nucleus occurs after liver resection. The IF is not so clear. The data would be far better when shown by compartment fractionation (nuclei/cytoplasm) in western blot.
- 4) In the same figure, the authors claim YAP localizes at the apical membrane after liver resection. It is not clear how the two phenomena - YAP shuttling to the nucleus and to the membrane - are correlated. Also, this doubt would be solved by compartment fractionation (nucleus, cytoplasm, membrane).
- 5) In figure 5 the authors claim YAP is localized at the apical membrane of primary hepatocytes. Looking at the IF pictures it is very hard to guess YAP subcellular distribution. There seems to be a problem with the specificity of the antibody. Again, biochemical analysis (fractionation and WB) or even z-stack reconstruction would help.
- 6) Figure 5b also misses statistical analysis, so it is not clear whether Y27 is actually able to reduce DCA effect. Same applies to Figure S7 and all the graphs of the manuscript.
- 7) The western blot in Figure 5c is quite obvious. Nonetheless, band quantification would make it self-explanatory.

Reviewer #2:

This is a wonderful manuscript that identifies physiological activation of YAP in response to bile

acids via apical domain expansion within bile canaliculi. The imaging data are very high quality and the overall model is well supported by the evidence. I have only minor concerns regarding the interpretation and presentation, but otherwise the manuscript should be published without delay.

Minor comments.

1. Are the authors aware that YAP responds to both mechanical stress (tension) and mechanical strain (stretching) of the apical domain of epithelial cells? See the Benham-Pyle et al Science or Fletcher et al Development papers for results on mechanical strain. The authors seem to have interpreted their findings solely through the lens of mechanical tension via Rho-kinase and p-MLC, but their findings are best explained by a combination of both tension and apical expansion regulating YAP. Obviously mechanical stress and strain occur simultaneously in this system, and the stress no doubt occurs in response to strain, but they are nevertheless distinct mechanisms.

2. Given the above comment, I found the emphasis on actomyosin a little misleading in the title and abstract. Why not re-title manuscript: "Expansion of bile canaliculi activates YAP during liver regeneration".

Reviewer #3:

In this manuscript, Meyer and colleagues described a new role of bile acids during liver regeneration. They showed that mice subjected to partial hepatectomy as well as bile duct ligation exhibit alteration of the bile acid canaliculi/duct network, with larger canaliculi associated with remodeling of the apical actomyosin cytoskeleton of hepatocytes, including enhanced phosphomyosin. They correlate this with YAP activation in the injured liver, which coincides with remodeling of bile canaliculi. During this time-window, they also observed that YAP localizes to the apical region of hepatocytes. In vitro, treatment of primary hepatocytes with bile acid (DCA) is sufficient to increase YAP nuclear localization and phosphorylation of MLC. To understand what is upstream of what, they combine DCA with drugs that inhibit MLC or the cytoskeleton, and show that only ROCK inhibition is capable of counteracting DCA-induced YAP localization, while this is not observed by treatment with Formin or F-actin inhibitors. According to these results, treatment of mice with Fasudil led to an even higher dilation of bile duct canaliculi, and normalizes YAP localization in mice with hepatectomy or bile duct ligation. Finally, the authors propose an *in silico* model that would explain how bile acid can regulate YAP.

The main message of the manuscript is that the liver senses its size after hepatectomy by a mechanism that depends on bile acids but not only on its receptor FXR: hepatectomy causes increased bile acids, that somehow activates MLC and thus YAP to induce hepatocyte differentiation and reconstitute a normal liver size.

This message has some problems:

- 1) the most upstream event is the dilation of bile duct canaliculi. In the text, the authors seem to relate this to BA overload observed upon hepatectomy. However, BA overload usually refers to accumulation of BA that occurs in the blood (secondary to reduced liver uptake?), and causes reduced liver bile synthesis in hepatocytes (<https://doi.org/10.1002/hep.26463>), and is different from cholestasis that induces retention of BA into the canaliculi. So, it remains hard to understand how dilation of bile duct canaliculi can occur as a direct consequence of BA overload in the blood.
- 2) recent data indicate that YAP and TAZ are marginally required for liver regeneration after hepatectomy (10.1038/emm.2017.205), with a mere 20% reduction of overall liver weight after two weeks. At a minimum, YAP/TAZ knockouts should become insensitive to accelerated regeneration triggered by bile acid supplementation (10.1126/science.1121435).
- 3) the authors imply that BA signals by increasing apical MLC and inhibiting the Hippo pathway, in some unknown manner that entails apical localization of YAP. This mechanism would be the unique novelty of this manuscript, but remains very poorly characterized and based on correlative data, also in light of previous literature. For example, the effect of BA on liver regeneration requires activation of FXR (10.1126/science.1121435), raising the issue of whether MLC is also a readout of FXR. Another possibility is that BA signal to the Hippo pathway indirectly, by altered expression of IQGAP, in turn impacting cell-cell adhesions and the Hippo pathway (10.1016/j.celrep.2013.10.030). Is this the case? A third possibility is that BA signal through

GPCRs, given that the BA membrane receptor TGR5 also plays a role in liver regeneration after hepatectomy (doi.org/10.1002/hep.26463), and given the known role of GPCRs as regulators of RHO-ROCK activity.

4) all papers dealing with mechanical regulation of YAP agree that inhibition of ROCK or NMII, in general, has weaker effects than complete disruption of F-actin. Here the contrary is true, because latrunculin and cytochalasin have no effects (they do not have any morphological effects either, raising the issue of what concentration was used, and whether the actin cytoskeleton was affected at all...). Given the very marginal effects on localization showed in the panels, a thorough analysis of multiple established YAP target genes (ANKRD1, BICC1, CYR61, TGFB2 etc.) would be important to complement these data. Anyway, even if this was the case, the authors must show that fasudil treatment impairs liver regeneration in general (i.e. quantification of liver weight, proliferation), and the effects of BA supplementation.

5) multiple evidence indicate that the regulation of YAP by F-actin occurs independently from the Hippo pathway. Here, the Hippo pathway is assumed to be at work (see the model in figure 7) without any experimental evidence. If the authors want to support this idea, they need to provide genetic data in support.

1st Revision - authors' response

3rd Dec 19

Following the Reviewers' requests, we added a significant amount of experimental data that provide further evidence in favour of our conclusions. In particular, 1) we added subcellular fractionation data to corroborate the increase in nuclear YAP during regeneration, 2) we performed co-immunoprecipitation and mass spectrometry analysis to identify YAP-interactors associated with the actin cytoskeleton, 3) demonstrated that actin drugs modulate YAP target gene expression and 4) extensively edited the text to clarify misunderstandings and improve the clarity of the text.

Below please find our point-by-point response.

Reviewer #1:

The manuscript by Meyer and collaborators addresses the topic of mechanosensing in liver by adopting quite a unique and original approach.

The experimental design is extremely neat and clear, the results are very interesting and potentially groundbreaking.

The conclusions, though, are not always well supported by the results shown.

The authors rely mostly on high resolution microscopy, where YAP visualisation is suboptimal. Biochemical experiments would highly increase the impact of the manuscript.

Briefly, the authors show bile canaliculi (BC) undergo dilation and branching right after hepatectomy and reverse at 3.5 days PH.

By high resolution microscopy, they measure the changes in BC perimeter and total membrane length, they show both increased to a similar extent and conclude that the phenomenon is due to an increase in total apical membrane of hepatocytes.

They hypothesise that this is a response to increased BA levels to induce liver growth during regeneration and found it is associated with a concomitant increase in apical acto-myosin contractility.

Since YAP is downstream of acto-myosin stress, the authors proceed to demonstrate that BA accumulation causes an activation of YAP in hepatocytes, thus causing their proliferation during regeneration.

The main result, which could be potentially groundbreaking, is that YAP localizes to the apical membrane in actin-rich fibers. In fact, so far no direct evidence of YAP being in direct contact with the cytoskeleton was given in vivo. This result probably deserves some deeper characterization. Also, a closer understanding of how this process is correlated with YAP nuclear translocation is probably needed. This is not clear at all.

Then the authors move to in vitro primary hepatocytes culture to show bile acid activates YAP through acto-myosin axis. This result was already proven in tumor cells (doi:10.1126/science.aav0173).

Finally, the manuscript proposes a model of YAP activation during liver regeneration.

In this reviewer's opinion, the manuscript is definitely of interest for the scientific community, but additional experiments would be needed to strengthen the results.

Major concerns:

1) Figs.1d, e lack the statistical analysis. This way it is not clear how significant the changes are.

The statistical analysis for Fig.1d, e is described in the Figure legend and shows that both the BC perimeter and membrane length are significantly increased after PH when compared to the untreated condition.

2) Results in figure 2 are not so striking. I am not sure this is enough to say there is an increased actomyosin contractility.

The apical F-actin levels may appear not so striking. However, note that whereas the sham operated mice show a 30% decrease of apical F-actin compared to untreated, regenerating livers have significantly higher F-actin levels. The decrease after sham operation is likely a result of the laparotomy, anesthesia and/or analgesia as described in the Results section (lines 114-119). The results demonstrate the importance of proper controls to account for side effects from surgical procedures. However, the quantification of apical pMLC levels (Fig.2d) shows a striking increase during regeneration. The combination of the two results justify our conclusion that acto-myosin contractility increases during regeneration.

3) In figure 3 and S4, the authors claim YAP translocation to the nucleus occurs after liver resection. The IF is not so clear. The data would be far better when shown by compartment fractionation (nuclei/cytoplasm) in western blot.

To verify nuclear YAP enrichment during regeneration, we performed nuclear fractionations from untreated, sham operated and regenerating livers (2 days post PH) and detected YAP protein in the cytosolic and nuclear and membrane fraction by Western blot. The results are shown in the new Fig.3f,g. Consistent with the IF data in Fig. 3, we find an increase of YAP in the nuclear fraction when compared to sham operated mice. These results are therefore in agreement with the IF results.

4) In the same figure, the authors claim YAP localizes at the apical membrane after liver resection. It is not clear how the two phenomena - YAP shuttling to the nucleus and to the membrane - are correlated. Also, this doubt would be solved by compartment fractionation (nucleus, cytoplasm, membrane).

How YAP shuttling to the nucleus is related to the apical membrane compartment is an interesting but difficult question. The temporal co-variation of apical and nuclear YAP does suggest that the two events are mechanistically linked. As the Reviewer requested, we performed membrane fractionation of regenerating livers (see answer to point 3) and detected YAP in the membrane and the nuclear fraction (Fig.3f,g). However, this type of approach does not have the resolution to infer causality and provide further insights into the regulatory function of the apical compartment on YAP. Also, we do not mean that YAP is necessarily associated with the membrane, we think that it is likely associated with the actin cortical mesh, and modified the text to make it clear. Indeed, we performed a YAP co-immunoprecipitation from regenerating livers to identify YAP interactors (see also response to Rev#3, point #3). Consistent with our hypothesis, we found an enrichment of actin cytoskeleton-associated proteins as well as a large set of tight junction proteins as YAP interactors and show these in Fig. S8 and Table S1-3. These candidate regulators provide a basis for a future dissection of apical YAP regulation.

5) In figure 5 the authors claim YAP is localized at the apical membrane of primary hepatocytes. Looking at the IF pictures it is very hard to guess YAP subcellular distribution. There seems to be a problem with the specificity of the antibody. Again, biochemical analysis (fractionation and WB) or even z-stack reconstruction would help.

We really cannot follow the Reviewer's criticism. First, we verified the specificity of the YAP antibody by showing the loss of YAP staining upon YAP silencing in liver tissue (Fig. S3). Second, we support our observations that YAP localizes to the apical compartment of hepatocytes and, in particular to F-actin rich structures, using immuno-EM and CLEM. To satisfy the Reviewer's request, we now performed fractionation of liver tissue from WT and regenerating livers showing that YAP is in the nuclear as well as membrane (100,000g) fraction. The membrane fraction does however contain insoluble F-actin and could reflect the association of YAP with the actin cytoskeleton (Fig.3f,g).

6) Figure 5b also misses statistical analysis, so it is not clear whether Y27 is actually able to reduce DCA effect. Same applies to Figure S7 and all the graphs of the manuscript.

All graphs of the manuscript have statistical analysis that are stated in the Figure legends. This includes Figure 5b which shows that Y27 significantly reduces the effect of DCA ($p = 0.05$). To make the statistical results easier to grasp we also added them to all bar graphs in the Figure panels as asterisks (Fig. 1d,e, Fig. 5b,e; Fig. S5c, Fig. S7a,b,c,d).

7) The western blot in Figure 5c is quite obvious. Nonetheless, band quantification would make it self-explanatory.

The quantification of the Western blot and additional independent experiments of Fig. 5c was already provided in Supp Fig. S7b.

Reviewer #2:

This is a wonderful manuscript that identifies physiological activation of YAP in response to bile acids via apical domain expansion within bile canaliculi. The imaging data are very high quality and the overall model is well supported by the evidence. I have only minor concerns regarding the interpretation and presentation, but otherwise the manuscript should be published without delay.

Minor comments.

1. Are the authors aware that YAP responds to both mechanical stress (tension) and mechanical strain (stretching) of the apical domain of epithelial cells? See the Benham-Pyle et al Science or Fletcher et al Development papers for results on mechanical strain. The authors seem to have interpreted their findings solely through the lens of mechanical tension via Rho-kinase and p-MLC, but their findings are best explained by a combination of both tension and apical expansion regulating YAP. Obviously mechanical stress and strain occur simultaneously in this system, and the stress no doubt occurs in response to strain, but they are nevertheless distinct mechanisms.

We completely agree with the Reviewer. Mechanical strain initially was not modeled explicitly and both stimuli were subsumed under one variable as both stimuli are proportional (pg. 19, line 94 of the Supplementary file has $\gamma_{(stress)}(x) \sim \frac{a(x)}{a_0(x)}$).

We now consider it explicitly. This mechanical strain of each hepatocyte's apical membrane can be modeled as the ratio of half the BC circumferences upon PH and the control condition

$$\gamma_{strain}(x) = \frac{2\pi a(x)/2}{2\pi a_0(x)/2} = \frac{a(x)}{a_0(x)} .$$

As both contributions are proportional to $a(x)/a_0(x)$ and any pre-factors are constants, the results remain unchanged up to a constant scaling factor of the dimensionless stimulus axis if specific weights in a linear combination of stress and strain are considered. We have rewritten the Supplemental Material (pg.20-21) accordingly and cite the relevant references.

2. Given the above comment, I found the emphasis on actomyosin a little misleading in the title and abstract. Why not re-title manuscript: "Expansion of bile canaliculi activates YAP during liver regeneration".

In our opinion, the generalization to “expansion” reduces the mechanistic content, neglecting the actin-dependency which is re-enforced by the IP data.

Other possibilities would be:

1. Bile canaliculi remodeling activates YAP via the acto-myosin system during liver regeneration
2. Acto-myosin dependent activation of YAP by bile canaliculi expansion during liver regeneration.

Reviewer #3:

In this manuscript, Meyer and colleagues described a new role of bile acids during liver regeneration. They showed that mice subjected to partial hepatectomy as well as bile duct ligation exhibit alteration of the bile acid canaliculi/duct network, with larger canaliculi associated with remodeling of the apical actomyosin cytoskeleton of hepatocytes, including enhanced phospho-myosin. They correlate this with YAP activation in the injured liver, which coincides with remodeling of bile canaliculi. During this time-window, they also observed that YAP localizes to the apical region of hepatocytes. In vitro, treatment of primary hepatocytes with bile acid (DCA) is sufficient to increase YAP nuclear localization and phosphorylation of MLC. To understand what is upstream of what, they combine DCA with drugs that inhibit MLC or the cytoskeleton, and show that only ROCK inhibition is capable of counteracting DCA-induced YAP localization, while this is not observed by treatment with Formin or F-actin inhibitors. According to these results, treatment of mice with Fasudil led to an even higher dilation of bile duct canaliculi, and normalizes YAP localization in mice with hepatectomy or bile duct ligation. Finally, the authors propose an in silico model that would explain how bile acid can regulate YAP.

The main message of the manuscript is that the liver senses its size after hepatectomy by a mechanism that depends on bile acids but not only on its receptor FXR: hepatectomy causes increased bile acids, that somehow activates MLC and thus YAP to induce hepatocyte differentiation and reconstitute a normal liver size. This message has some problems:

1) the most upstream event is the dilation of bile duct canaliculi. In the text, the authors seem to relate this to BA overload observed upon hepatectomy. However, BA overload usually refers to accumulation of BA that occurs in the blood (secondary to reduced liver uptake?), and causes reduced liver bile synthesis in hepatocytes (<https://doi.org/10.1002/hep.26463>), and is different from cholestasis that induces retention of BA into the canaliculi. So, it remains hard to understand how dilation of bile duct canaliculi can occur as a direct consequence of BA overload in the blood.

It is widely accepted in the field that partial hepatectomy not only causes an increase in serum bile acid levels but also a concomitant increase of bile acids in liver tissue as a consequence of the enterohepatic circulation, (see e.g. Pean et al., *Hepatology*, 2013; Ding et al., *Molecular Medicine Reports*, 2015; Uriarte et al., *Gut*, 2013). Similarly, BA administered intravenously in absence of hepatectomy (e.g. Klaassen 1974) caused a corresponding increase in bile fluid formation. This shows that higher BA load in blood does stimulate hepatic transport and secretion

of BA and other osmolytes into BC, consequently increasing the BC osmotic pressure, water inflow and fluid pressure.

We agree that bile acid overload and cholestasis are perturbations of bile acid homeostasis resulting from different causes (increased BA in the entero-hepatic circulation and perturbation of bile flow, respectively). Yet, the physiological problem is similar – the liver experiences an increased bile acid load which, due to the toxicity of bile acids, needs to be resolved. Our observed dilation of the BC network upon both PH and cholestasis is thus likely a direct consequence of such overload to avoid hepatic toxicity from elevated BA levels in the liver.

2) recent data indicate that YAP and TAZ are marginally required for liver regeneration after hepatectomy (10.1038/emm.2017.205), with a mere 20% reduction of overall liver weight after two weeks. At a minimum, YAP/TAZ knockouts should become insensitive to accelerated regeneration triggered by bile acid supplementation (10.1126/science.1121435).

There may be a misunderstanding. Our aim was not to study the requirement of YAP/TAZ in liver regeneration but to use it only as a marker of a signaling pathway that responds to 1) bile acids and 2) mechanical signals during liver regeneration (Results pg.6, line 128-134). Our study provides a new answer to the question of how YAP responds to BA.

We agree that the previously reported impact of YAP/TAZ knockout on liver regeneration is smaller than one might expect (~20% reduction of liver mass; Lu et al., Exp and Mol Medicine, 2018) given the dramatic effects of Hippo signaling perturbations under homeostatic conditions (Zhou et al., Cancer Cell, 2009; Lu et al., PNAS 2010; Song et al., PNAS, 2010). The cited study did provide evidence that “*Yap/Taz-depleted livers exhibit profound defects in liver regeneration*”. Therefore, using YAP/TAZ as marker is not unjustified. We clarified this point in the text. More importantly, the precise requirement of YAP/TAZ signaling during liver regeneration is unclear: To the best of our knowledge, the YAP/TAZ knockout phenotype has only been assessed with respect to hepatocyte proliferation. Yet, YAP is also a regulator of hepatocyte differentiation (Yimlamai et al., Cell, 2014; Fitamant et al., Cell Report, 2015), ploidy (Zhang et al., Cancer Cell, 2017), identity/metabolic zonation (Fitamant et al., Cell Report, 2015) and cell size (Mugahid et al., BioRxiv, 2018; Tumaneng et al., Nat. Cell Biol., 2012). The YAP/TAZ knockout study lacks a detailed analysis of the cell type that proliferate to compensate tissue resection (stem cells or hepatocytes), and increase in cell size as an alternative way to increase liver mass as well as ploidy and tissue function. Thus, the function of YAP/TAZ during liver regeneration is only marginally understood to be able to dismiss it. However, although this is an important question, the Reviewer will agree that, given the premise of our study (not to focus on Hippo function in liver regeneration *per se* but use it as readout of BA response and mechanosensing), performing a complete analysis of conditional YAP/TAZ knockout mice (YAP KO is lethal and needs a conditional KO) would be a full study beyond the scope of this manuscript.

3) the authors imply that BA signals by increasing apical MLC and inhibiting the Hippo pathway, in some unknown manner that entails apical localization of YAP. This mechanism would be the unique novelty of this manuscript, but remains very poorly characterized and based on correlative data, also in light of previous

literature. For example, the effect of BA on liver regeneration requires activation of FXR (10.1126/science.1121435), raising the issue of whether MLC is also a readout of FXR. Another possibility is that BA signal to the Hippo pathway indirectly, by altered expression of IQGAP, in turn impacting cell-cell adhesions and the Hippo pathway (10.1016/j.celrep.2013.10.030). Is this the case? A third possibility is that BA signal through GPCRs, given that the BA membrane receptor TGR5 also plays a role in liver regeneration after hepatectomy (doi.org/10.1002/hep.26463), and given the known role of GPCRs as regulators of RHO-ROCK activity.

To gain insight into the potential molecular components involved in this system and further verify that YAP interacts with the actin cytoskeleton and/or associated proteins, we performed YAP immunoprecipitation and mass spectrometric analysis of YAP interactors from liver tissue at 1.5d post PH (Fig. S8, Table S1-3). Consistent with our results, we found a significant enrichment of actin cytoskeleton associated proteins as well as a large set of tight junction proteins among YAP interactors. These results support the hypothesis that YAP is regulated through a mechanism that senses mechanical or morphological cellular properties via the actin cytoskeleton. Based on the apical localization of YAP, these effects are likely caused by BA-induced BC network remodeling during regeneration. However, to fully answer the Reviewer's question, one would need a dedicated project on the function of the YAP interactors with respect to the link between BAs and BC mechanics/structure.

4) all papers dealing with mechanical regulation of YAP agree that inhibition of ROCK or NMII, in general, has weaker effects than complete disruption of F-actin. Here the contrary is true, because latrunculin and cytochalasin have no effects (they do not have any morphological effects either, raising the issue of what concentration was used, and whether the actin cytoskeleton was affected at all...). Given the very marginal effects on localization showed in the panels, a thorough analysis of multiple established YAP target genes (ANKRD1, BICCI1, CYR61, TGFB2 etc.) would be important to complement these data. Anyway, even if this was the case, the authors must show that fasudil treatment impairs liver regeneration in general (i.e. quantification of liver weight, proliferation), and the effects of BA supplementation.

Figure 5 is very complex and we understand that details can easily be overlooked (which we tried to improve). However, we respectfully disagree with the Reviewer's statement that CytoD has no apparent effect on F-actin or YAP. Fig.5d shows a dramatic dilation of bile canaliculi upon CytoD treatment. BC are about 2-3 μm in control conditions while they expand to the size of nuclei (!) upon CytoD treatment. Furthermore, CytoD causes a clear fragmentation of F-actin as shown by Phalloidin staining (see insets of the Figure). The phenotype indicates that the drug concentration is sufficient for a strong perturbation of the actin cytoskeleton. To make these morphological alterations clearer, we marked them with arrows/asterisks in the figure panel and explained them in the legend.

Interestingly and in contrast to previous studies, we observe that LatA and CytoD cause an increase of nuclear YAP instead of a depletion. This is in contrast to the acto-myosin inhibitor Y27 which decreases nuclear YAP (Fig.5e). To verify the

activation of YAP and as requested by the Reviewer, we performed qPCR analysis of YAP target genes (Ankrd1, Cyr61, CTGF) in primary hepatocyte culture upon actin inhibitor treatment. The new results (pg. 10, lines 236-238, Fig. S7d) are fully consistent with our observations demonstrating that Y27-mediated nuclear YAP depletion decreases YAP target gene expression, whereas CytoD-mediated increase of nuclear YAP induces YAP target gene expression.

The Reviewer's additional request to study the effect of Fasudil on hepatocyte proliferation during liver regeneration is unfortunately not feasible. YAP is activated early on during regeneration but hepatocyte proliferation only peaks at about 2d. Thus, it would require the treatment of mice with Fasudil for multiple days, requiring many injections to maintain inhibitor concentrations sufficiently high in the liver. As such this experiment is not justifiable. The alternative genetic ablation of RhoA might be a way to address this point. Since primary hepatocytes barely proliferate in culture, this requires the study of liver regeneration in RhoA-knockout mouse which goes beyond the scope of this study.

5) multiple evidence indicate that the regulation of YAP by F-actin occurs independently from the Hippo pathway. Here, the Hippo pathway is assumed to be at work (see the model in figure 7) without any experimental evidence. If the authors want to support this idea, they need to provide genetic data in support.

As pointed out above, we are using Hippo and YAP/TAZ as a marker of actomyosin readout in signaling. It is not our intention to study the requirement of Hippo *per se* during liver regeneration but to understand how it is activated by BA during liver regeneration. The sole purpose of our mathematical model is to provide a general theoretical framework for the regulation of signaling pathways (e.g. YAP) by a mechanical stimulus during liver regeneration. The model shares structural aspects with the Hippo pathway but that does not mean that the Hippo pathway is the only responsible. Rather it shows that simple mass action and Michaelis-Menten kinetics of a pathway with similar topology as found in the Hippo pathway (with an amplifying cascade and cytosolic sequestration) is sufficient to describe the observed behavior. This should help the future identification of the missing links between F-actin and YAP, or other signaling pathway that must be activated and switched off when liver regeneration is complete. Since the way the text was written and the model was presented have caused confusion about the involvement of the Hippo pathway kinases, we rephrased the text (pg. 12-13, lines 286-300) and changed the model nomenclature of the model variables in Fig. 7a to avoid misunderstandings.

In summary, we thank the Reviewers for their comments and guidance, and are very pleased with the resulting improvements. Having responded to the Reviewers' comments, we hope that the manuscript can now be accepted for publication.

Thank you for sending us your revised manuscript. We have now heard back from the two reviewers who were asked to evaluate your study. Overall, the reviewers think that the study has improved as a result of the performed revisions. However, as you will see below, reviewer #3 still raises some remaining concerns. We think that *in vivo* experiments in YAP/TAZ knockout mice and a precise mechanistic dissection of the role of YAP in the process are not mandatory for the acceptance of the study. We also consulted with reviewer #1 regarding these concerns and they still felt that the study is suitable for publication. As such, we would only ask you to perform minor revisions, providing some final clarifications regarding the remaining points (2, 3 and 4) of reviewer #3.

REFEREE REPORTS

Reviewer #1:

The authors clarified all the doubts I had and included significant experiments to the new version of the manuscript. It is my belief the manuscript is now stronger and suitable for publication in the present form.

Reviewer #3:

In this revised version of the manuscript, the authors do not provide substantial advancements, despite they could have well attempted at least some of the requested experiments. For example, the authors had liver YAP KO available to perform control IF stainings (see below), but the requested experiments on regeneration were skipped. Similarly, treatment of mice with ROCK inhibitor in drinking water is very easy to do (see below), but the authors did not do the experiment. The authors state these data are beyond the scope of the article, because they are using YAP/TAZ as read-outs for a pathway that responds to bile acids and to mechanical signals. However, the data the authors provide is different or opposite to the literature where the link between YAP/TAZ and bile acids / mechanics was established. So this is not a marker, but a novel pathway and a novel mechanism (as also claimed in the title), which however lacks of any characterization and, perhaps more importantly, of any biological relevance in the context of liver regeneration.

1) the most upstream event is the dilation of bile duct canaliculi. In the text, the authors seem to relate this to BA overload observed upon hepatectomy. However, BA overload usually refers to accumulation of BA that occurs in the blood (secondary to reduced liver uptake?), and causes reduced liver bile synthesis in hepatocytes (<https://doi.org/10.1002/hep.26463>), and is different from cholestasis that induces retention of BA into the canaliculi. So, it remains hard to understand how dilation of bile duct canaliculi can occur as a direct consequence of BA overload in the blood.

authors' response to 1) It is widely accepted in the field that partial hepatectomy not only causes an increase in serum bile acid levels but also a concomitant increase of bile acids in liver tissue as a consequence of the enterohepatic circulation, (see e.g. Pean et al., *Hepatology*, 2013; Ding et al., *Molecular Medicine Reports*, 2015; Uriarte et al., *Gut*, 2013). Similarly, BA administered intravenously in absence of hepatectomy (e.g. Klaassen 1974) caused a corresponding increase in bile fluid formation. This shows that higher BA load in blood does stimulate hepatic transport and secretion of BA and other osmolytes into BC, consequently increasing the BC osmotic pressure, water inflow and fluid pressure. We agree that bile acid overload and cholestasis are perturbations of bile acid homeostasis resulting from different causes (increased BA in the entero-hepatic circulation and perturbation of bile flow, respectively). Yet, the physiological problem is similar - the liver experiences an increased bile acid load which, due to the toxicity of bile acids, needs to be resolved. Our observed dilation of the BC network upon both PH and cholestasis is thus likely a direct consequence of such overload to avoid hepatic toxicity from elevated BA levels in the liver.

revision of point 1) Thanks for the clarification.

2) recent data indicate that YAP and TAZ are marginally required for liver regeneration after hepatectomy (10.1038/emmm.2017.205), with a mere 20% reduction of overall liver weight after two weeks. At a minimum, YAP/TAZ knockouts should become insensitive to accelerated regeneration triggered by bile acid supplementation (10.1126/science.1121435).

authors' response to 2) There may be a misunderstanding. Our aim was not to study the requirement of YAP/TAZ in liver regeneration but to use it only as a marker of a signaling pathway that responds to 1) bile acids and 2) mechanical signals during liver regeneration (Results pg.6, line 128-134). Our study provides a new answer to the question of how YAP responds to BA. We agree that the previously reported impact of YAP/TAZ knockout on liver regeneration is smaller than one might expect (~20% reduction of liver mass; Lu et al., *Exp and Mol Medicine*, 2018) given the dramatic effects of Hippo signaling perturbations under homeostatic conditions (Zhou et al., *Cancer Cell*, 2009; Lu et al., *PNAS* 2010; Song et al., *PNAS*, 2010). The cited study did provide evidence that "Yap/Taz-depleted livers exhibit profound defects in liver regeneration". Therefore, using YAP/TAZ as marker is not unjustified. We clarified this point in the text. More importantly, the precise requirement of YAP/TAZ signaling during liver regeneration is unclear: To the best of our knowledge, the YAP/TAZ knockout phenotype has only been assessed with respect to hepatocyte proliferation. Yet, YAP is also a regulator of hepatocyte differentiation (Yimlamai et al., *Cell*, 2014; Fitamant et al., *Cell Report*, 2015), ploidy (Zhang et al., *Cancer Cell*, 2017), identity/metabolic zonation (Fitamant et al., *Cell Report*, 2015) and cell size (Mugahid et al., *BioRxiv*, 2018; Tumaneng et al., *Nat. Cell Biol.*, 2012). The YAP/TAZ knockout study lacks a detailed analysis of the cell type that proliferate to compensate tissue resection (stem cells or hepatocytes), and increase in cell size as an alternative way to increase liver mass as well as ploidy and tissue function. Thus, the function of YAP/TAZ during liver regeneration is only marginally understood to be able to dismiss it. However, although this is an important question, the Reviewer will agree that, given the premise of our study (not to focus on Hippo function in liver regeneration per se but use it as readout of BA response and mechanosensing), performing a complete analysis of conditional YAP/TAZ knockout mice (YAP KO is lethal and needs a conditional KO) would be a full study beyond the scope of this manuscript.

revision of point 2) The review understands the idea of using YAP/TAZ as a marker for bile acids signaling and/or mechanical perturbations of the liver tissue. However, the authors do not just use YAP/TAZ as markers, but use YAP/TAZ data to infer the existence of a novel pathway which is different from what previously published on bile acids and mechanical regulations. So, the issues of whether this pathway really works as the authors claim, and of whether this is relevant for the main biological process on which the study is focused, is very well justified. If YAP/TAZ are not required for regeneration in the experimental conditions used here, then what is the relevance of this novel type of YAP/TAZ regulation?

Please also note that, while in general it is true that asking for genetic data can be unfair given the time it takes to implement this type of experiments, in Fig S3 the authors do use YAP knockouts to control for the specificity of the antibody. So, they had the system available to perform the requested experiments, and to know whether or not the proposed pathway is functionally relevant.

3) the authors imply that BA signals by increasing apical MLC and inhibiting the Hippo pathway, in some unknown manner that entails apical localization of YAP. This mechanism would be the unique novelty of this manuscript, but remains very poorly characterized and based on correlative data, also in light of previous literature. For example, the effect of BA on liver regeneration requires activation of FXR (10.1126/science.1121435), raising the issue of whether MLC is also a readout of FXR. Another possibility is that BA signal to the Hippo pathway indirectly, by altered expression of IQGAP, in turn impacting cell-cell adhesions and the Hippo pathway (10.1016/j.celrep.2013.10.030). Is this the case? A third possibility is that BA signal through GPCRs, given that the BA membrane receptor TGR5 also plays a role in liver regeneration after hepatectomy (doi.org/10.1002/hep.26463), and given the known role of GPCRs as regulators of RHO-ROCK activity.

authors' response to 3) To gain insight into the potential molecular components involved in this system and further verify that YAP interacts with the actin cytoskeleton and/or associated proteins, we performed YAP immunoprecipitation and mass spectrometric analysis of YAP interactors from

liver tissue at 1.5d post PH (Fig. S8, Table S1-3). Consistent with our results, we found a significant enrichment of actin cytoskeleton associated proteins as well as a large set of tight junction proteins among YAP interactors. These results support the hypothesis that YAP is regulated through a mechanism that senses mechanical or morphological cellular properties via the actin cytoskeleton. Based on the apical localization of YAP, these effects are likely caused by BA-induced BC network remodeling during regeneration. However, to fully answer the Reviewer's question, one would need a dedicated project on the function of the YAP interactors with respect to the link between BAs and BC mechanics/structure.

revision of point 3) The interaction with YAP/TAZ with cytoskeletal-associated proteins or has been reported in many studies, but on its own this cannot suffice to "support the hypothesis that YAP is regulated through a mechanism that senses mechanical or morphological cellular properties via the actin cytoskeleton". The authors propose here a different observation and mechanisms compared to previous work focusing on this topic, but it remains unclear how the simple identification of a collection of potential YAP/TAZ interactors and potential cytoskeletal interactors may shed light on how this novel regulation occurs.

Moreover, the main request here was to understand better how BA signaling connects with YAP/TAZ in the context of liver regeneration, taking into account multiple papers that already approached segments of this process in details, and that were based on strong evidence. The authors now propose a mechanism which is different from what is already known, but do not provide experimental data to support their view. At least, it would have been worth trying to reconcile these different views.

4) all papers dealing with mechanical regulation of YAP agree that inhibition of ROCK or NMII, in general, has weaker effects than complete disruption of F-actin. Here the contrary is true, because latrunculin and cytochalasin have no effects (they do not have any morphological effects either, raising the issue of what concentration was used, and whether the actin cytoskeleton was affected at all...). Given the very marginal effects on localization showed in the panels, a thorough analysis of multiple established YAP target genes (ANKRD1, BICC1, CYR61, TGFB2 etc.) would be important to complement these data. Anyway, even if this was the case, the authors must show that fasudil treatment impairs liver regeneration in general (i.e. quantification of liver weight, proliferation), and the effects of BA supplementation.

authors' response to 4) Figure 5 is very complex and we understand that details can easily be overlooked (which we tried to improve). However, we respectfully disagree with the Reviewer's statement that CytoD has no apparent effect on F-actin or YAP. Fig.5d shows a dramatic dilation of bile canaliculi upon CytoD treatment. BC are about 2-3 μ m in control conditions while they expand to the size of nuclei (!) upon CytoD treatment. Furthermore, CytoD causes a clear fragmentation of F-actin as shown by Phalloidin staining (see insets of the Figure). The phenotype indicates that the drug concentration is sufficient for a strong perturbation of the actin cytoskeleton. To make these morphological alterations clearer, we marked them with arrows/asterisks in the figure panel and explained them in the legend. Interestingly and in contrast to previous studies, we observe that Lata and CytoD cause an increase of nuclear YAP instead of a depletion. This is in contrast to the actomyosin inhibitor Y27 which decreases nuclear YAP (Fig.5e). To verify the activation of YAP and as requested by the Reviewer, we performed qPCR analysis of YAP target genes (Ankrd1, Cyr61, CTGF) in primary hepatocyte culture upon actin inhibitor treatment. The new results (pg. 10, lines 236-238, Fig. S7d) are fully consistent with our observations demonstrating that Y27-mediated nuclear YAP depletion decreases YAP target gene expression, whereas CytoD-mediated increase of nuclear YAP induces YAP target gene expression. The Reviewer's additional request to study the effect of Fasudil on hepatocyte proliferation during liver regeneration is unfortunately not feasible. YAP is activated early on during regeneration but hepatocyte proliferation only peaks at about 2d. Thus, it would require the treatment of mice with Fasudil for multiple days, requiring many injections to maintain inhibitor concentrations sufficiently high in the liver. As such this experiment is not justifiable. The alternative genetic ablation of RhoA might be a way to address this point. Since primary hepatocytes barely proliferate in culture, this requires the study of liver regeneration in RhoA-knockout mouse which goes beyond the scope of this study.

revision of point 4) Contrary to what the authors claim above on the use of YAP/TAZ as mere markers, the authors do present a new mechanism of regulation of YAP/TAZ which is different

from previous literature. If one uses a marker, it must be in the same direction of the published use of this marker. Otherwise it is a novel observation, which needs explanations.

The experiment provided in S7d has serious problems: (i) it shows that treatment of hepatocytes with the Y27632 inhibitor inhibits Ankrd1, Ctgf and Cyr61, however this is not recapitulated by treatment with Fasudil, which is an independent small-molecule inhibitor of ROCK (at an unknown dose). In principle, this raises serious doubts on the specificity of the Y27632, which is known to have potential off-target effects like any other kinase inhibitor, which will be different from the off-target effects of Fasudil. (ii) it does show that latrunculinA inhibits the target genes, which is contrary to the immunolocalization results on YAP. How do the authors explain this? Very likely, the authors are not considering the existence of other pathways regulated by actin. The effects of latrunculinA are fully compatible with inhibition of both YAP/TAZ but also of MAL/SRF transcription factors. Indeed it is known from the literature that the Ankrd1, Ctgf and Cyr61 are all direct target genes of MAL/SRF complexes (Foster G&D 2017). Several other more specific YAP/TAZ target genes are known in hepatocytes, given the extensive literature on Hippo and liver. Moreover, this is also perfectly in line with the results with cytochalasinD, which kills F-actin but activates MAL/SRF (see Miralles Cell 2003), and coherently induces activation of the three target genes. At a minimum, experimental data should have been provided that these regulations depend on YAP/TAZ based on their knockdown. Thus, in the end it is hard to accept that this experiment can be used to reinforce the notion that YAP/TAZ are regulated in a coherent manner with the author's claims.

Finally, the authors also avoided to perform the requested experiment with ROCK-inhibitor treatment. They claim it is not possible because it would require multiple daily injections. However, orally-available formulations of Fasudil are available, and this can be prolonged for long periods with effects on multiple organs including the liver (10.3389/fphar.2017.00017, 10.1161/01.CIR.0000163544.17221.BE, 10.1161/01.HYP.0000221605.94532.71, 10.1111/bph.12277, 10.1371/journal.pone.0110446). So the experiment was well feasible.

5) multiple evidence indicate that the regulation of YAP by F-actin occurs independently from the Hippo pathway. Here, the Hippo pathway is assumed to be at work (see the model in figure 7) without any experimental evidence. If the authors want to support this idea, they need to provide genetic data in support.

authors' response to 5) As pointed out above, we are using Hippo and YAP/TAZ as a marker of actomyosin readout in signaling. It is not our intention to study the requirement of Hippo per se during liver regeneration but to understand how it is activated by BA during liver regeneration. The sole purpose of our mathematical model is to provide a general theoretical framework for the regulation of signaling pathways (e.g. YAP) by a mechanical stimulus during liver regeneration. The model shares structural aspects with the Hippo pathway but that does not mean that the Hippo pathway is the only responsible. Rather it shows that simple mass action and Michaelis-Menten kinetics of a pathway with similar topology as found in the Hippo pathway (with an amplifying cascade and cytosolic sequestration) is sufficient to describe the observed behavior. This should help the future identification of the missing links between F-actin and YAP, or other signaling pathway that must be activated and switched off when liver regeneration is complete. Since the way the text was written and the model was presented have caused confusion about the involvement of the Hippo pathway kinases, we rephrased the text (pg. 12-13, lines 286-300) and changed the model nomenclature of the model variables in Fig. 7a to avoid misunderstandings.

revision of point 5) Thanks for having clarified this.

2nd Revision - authors' response

19th Jan 20

I would like to thank you and the reviewers again for the positive response to our revised manuscript and are delighted that it is suitable for publication, pending minor revisions. Please find below our response to the remaining concerns of reviewer #3 points 2-4.

We edited the text of the manuscript to meet the reviewer's criticism and avoid misinterpretations about the role of acto-myosin in the regulation of YAP during regeneration (this was also requested by Reviewer#2 in the first revision). We additionally changed the title in order to tone down the role of the acto-myosin system (again in response to Reviewer 3).

Additionally, please find below our response to your question regarding the image integrity in Figure S5B. To meet the standards of MSB we further modified the Methods section, added p-values to all Figure legends, refer to Figure S8 in the main text and provide the model code. We added individual data points to graphs that have $n < 5$, however, this is not always possible. Fig. 1b and 3d would become unreadable if we added all datapoints. For Fig. 2b,d and 3c we simply cannot provide them due to the way the data were calculated: We took all raw data of time courses, calculated the average, normalized this to the WT and displayed the normalized values. Thus, we do not have the individual normalized datapoints. For all these cases we now provide the source data (raw, averaged and normalized data) to make these graphs as transparent as possible to the reader.

Reply to reviewer #3

Point 2. The review understands the idea of using YAP/TAZ as a marker for bile acids signaling and/or mechanical perturbations of the liver tissue. However, the authors do not just use YAP/TAZ as markers, but use YAP/TAZ data to infer the existence of a novel pathway which is different from what previously published on bile acids and mechanical regulations. So, the issues of whether this pathway really works as the authors claim, and of whether this is relevant for the main biological process on which the study is focused, is very well justified. If YAP/TAZ are not required for regeneration in the experimental conditions used here, then what is the relevance of this novel type of YAP/TAZ regulation?

Please also note that, while in general it is true that asking for genetic data can be unfair given the time it takes to implement this type of experiments, in Fig S3 the authors do use YAP knockouts to control for the specificity of the antibody. So, they had the system available to perform the requested experiments, and to know whether or not the proposed pathway is functionally relevant.

We do not disagree that the dissection of YAP function for liver regeneration is an important question. However, as already stated in the initial reply to this request, we do not have the required tools to dissect this genetically. The conditional YAP KO only generates a mosaic knockout upon virus induction (see Fig. S3, ~ 50% of hepatocytes show YAP depletion). This means that not all cells are equally affected and, importantly, the loss of YAP is very likely compensated by TAZ. The reported liver regeneration phenotype (Lu et al., 2018) was observed in YAP/TAZ double KO mice. This is why we maintain our opinion that elucidating YAP function for liver regeneration is a full study that goes beyond the scope of this (already extended) manuscript.

Point 3. The interaction with YAP/TAZ with cytoskeletal-associated proteins or has been reported in many studies, but on its own this cannot suffice to "support the hypothesis that YAP is regulated through a mechanism that senses mechanical or morphological cellular properties via the actin cytoskeleton". The authors propose here a different observation and mechanisms compared to previous work focusing

on this topic, but it remains unclear how the simple identification of a collection of potential YAP/TAZ interactors and potential cytoskeletal interactors may shed light on how this novel regulation occurs.

Moreover, the main request here was to understand better how BA signaling connects with YAP/TAZ in the context of liver regeneration, taking into account multiple papers that already approached segments of this process in details, and that were based on strong evidence. The authors now propose a mechanism which is different from what is already known, but do not provide experimental data to support their view. At least, it would have been worth trying to reconcile these different views.

If YAP/TAZ were regulated by the actin cytoskeleton in the liver one would expect it to interact with cytoskeletal components: The list of potential YAP/TAZ interactors supports this hypothesis. Clearly, this is only the beginning and a full functional study on the candidates identified is required to gain mechanistic insights. In general, we agree that it remains an open question how our observations and previous reports of liver regeneration and bile acid signaling (e.g. via FXR or IQGAP) act together. Are these different pathways or do nuclear receptors also signal through the actin cytoskeleton? We did however cite and discuss these reports (e.g. Huang et al, 2006 and Anakk et al. 2013) in the Discussion. We re-edited the text to emphasize this point further (lines 352-361).

Point 4. *Contrary to what the authors claim above on the use of YAP/TAZ as mere markers, the authors do present a new mechanism of regulation of YAP/TAZ which is different from previous literature. If one uses a marker, it must be in the same direction of the published use of this marker. Otherwise it is a novel observation, which needs explanations.*

The experiment provided in S7d has serious problems: (i) it shows that treatment of hepatocytes with the Y27632 inhibitor inhibits Ankrd1, Ctgf and Cyr61, however this is not recapitulated by treatment with Fasudil, which is an independent small-molecule inhibitor of ROCK (at an unknown dose). In principle, this raises serious doubts on the specificity of the Y27632, which is known to have potential off-target effects like any other kinase inhibitor, which will be different from the off-target effects of Fasudil. (ii) it does show that latrunculinA inhibits the target genes, which is contrary to the immunolocalization results on YAP. How do the authors explain this? Very likely, the authors are not considering the existence of other pathways regulated by actin. The effects of latrunculinA are fully compatible with inhibition of both YAP/TAZ but also of MAL/SRF transcription factors. Indeed it is known from the literature that the Ankrd1, Ctgf and Cyr61 are all direct target genes of MAL/SRF complexes (Foster G&D 2017). Several other more specific YAP/TAZ target genes are known in hepatocytes, given the extensive literature on Hippo and liver. Moreover, this is also perfectly in line with the results with cytochalasinD, which kills F-actin but activates MAL/SRF (see Miralles Cell 2003), and coherently induces activation of the three target genes. At a minimum, experimental data should have been provided that these regulations depend on YAP/TAZ based on their knockdown. Thus, in the end it is hard to accept that this experiment can be used to reinforce the notion that YAP/TAZ are regulated in a coherent manner with the author's claims.

Finally, the authors also avoided to perform the requested experiment with ROCK-inhibitor treatment. They claim it is not possible because it would require multiple daily injections. However, orally-available formulations of Fasudil are available, and this can be prolonged for long periods with effects on multiple organs including the liver (10.3389/fphar.2017.00017, 10.1161/01.CIR.0000163544.17221.BE, 10.1161/01.HYP.0000221605.94532.71, 10.1111/bph.12277, 10.1371/journal.pone.0110446). So the experiment was well feasible.

We fully understand and agree with the reviewer's criticism and invitation to caution with respect to interpretation of the mechanism. We certainly agree that the Hippo field in general has not elucidated the precise molecular mechanisms underlying the mechanosensing function of the actin cytoskeleton which is read out by YAP. We concur with the Reviewer that, although the actin inhibitors generally show modulatory effects on YAP, they do not always give a consistent response. Given the complexity of transcriptional regulation as noted by the Reviewer, this is even not too surprising. Therefore, we edited the text to point this out more clearly and avoid as much as possible over- and mis-interpretations. This includes the observations, that

1) LatA and CytoD have opposite effects on YAP target gene expression although both induce nuclear YAP levels and MLC phosphorylation (lines 243-248). As stated by the reviewer, other factors affected by the actin inhibitors (such as MAL/SRF) may confound gene expression readouts and target genes may be sensitive to different nuclear YAP levels or timings that we did not study. However, this also shows that the study of target genes upon strong perturbations such as the actin cytoskeleton are problematic to draw conclusions from and may not provide much further insights. We think that the YAP localization data upon actin perturbation *in vitro* and *in vivo* still provides strong evidence that YAP does sense a mechanical or morphological property of the BC network. Thus, we do not agree that the above described inconsistencies of the gene expression data argue against that.

2) Y27 and Fasudil, two inhibitors that target the same molecule (ROCK), differ in their strength of inhibition of myosin light chain phosphorylation and YAP (lines 257-261). We also added a reference on the distinct effects of the two inhibitors (Ichikawa et al. Brain Res. 2008).

The finding that inhibition of acto-myosin activity by Fasudil is sufficient to inhibit YAP nuclear localization during liver regeneration supports a regulation of YAP by the acto-myosin system, but does not exclude other mechanisms.

Altogether, in our study the drugs were used to establish a general correlation between acto-myosin regulation and YAP translocation, preluding to the validation using Fasudil *in vivo*. We found that bile acid- and PH-induced YAP activation depend at least in part on the actin cytoskeleton, specifically the acto-myosin system. However, we edited the text to also consider additional indirect or actin-independent mechanisms involved in the regulation of YAP upon BC remodeling, e.g. apical membrane size (BC dilate), membrane tension/osmotic pressure due to BA secretion, cell-cell junction integrity.

Accepted

23rd Jan 20

Thank you again for sending us your revised manuscript and for fixing all the remaining editorial issues. We are now satisfied with the modifications made and I am pleased to inform you that your paper has been accepted for publication.

Corresponding Author Name: Marino Zerial
Journal Submitted to: Molecular Systems Biology
Manuscript Number: MSB-19-8985R